# RNA m⁶A methylation modulates airway inflammation in allergic asthma via PTX3-dependent macrophage homeostasis

Xiao Han [1,2,7] ✉, Lijuan Liu[3,7], Saihua Huang[1,2,7], Wenfeng Xiao[1,2,7], Yajing Gao [1,2,7], Weitao Zhou[3], Caiyan Zhang[4], Hongmei Zheng[3], Lan Yang[1,2], Xueru Xie[1,2], Qiuyan Liang[1,2], Zikun Tu[1,2], Hongmiao Yu[1,2], Jinrong Fu [5], Libo Wang[3], Xiaobo Zhang[3], Liling Qian [3,6] ✉ & Yufeng Zhou [1,2] ✉

$N^6$-methyladenosine (m⁶A), the most prevalent mRNA modification, has an important function in diverse biological processes. However, the involvement of m⁶A in allergic asthma and macrophage homeostasis remains largely unknown. Here we show that m⁶A methyltransferases *METTL3* is expressed at a low level in monocyte-derived macrophages from childhood allergic asthma patients. Conditional knockout of *Mettl3* in myeloid cells enhances Th2 cell response and aggravates allergic airway inflammation by facilitating M2 macrophage activation. Loss and gain functional studies confirm that METTL3 suppresses M2 macrophage activation partly through PI3K/AKT and JAK/STAT6 signaling. Mechanistically, m⁶A-sequencing shows that loss of *METTL3* impairs the m⁶A-YTHDF3-dependent degradation of *PTX3* mRNA, while higher PTX3 expression positively correlates with asthma severity through promoting M2 macrophage activation. Furthermore, the METTL3/YTHDF3-m⁶A/PTX3 interactions contribute to autophagy maturation in macrophages by modulating *STX17* expression. Collectively, this study highlights the function of m⁶A in regulating macrophage homeostasis and identifies potential targets in controlling allergic asthma.

Asthma is a heterogeneous disease characterized by bronchial hyper-responsiveness, airway remodeling, and chronic inflammation[1]. Allergic asthma is the most common clinical phenotypes of asthma. Notably, most school-age children have allergic asthma, which has often obvious involvement in the immune system such as eosinophils and type 2 helper T cells (Th2 cells)[2]. Children with allergic asthma have

concomitant allergic sensitization, which has been associated with asthma inception and severity[3]. Allergens from house dust mites, cockroaches, and furred pets represent the most common indoor triggers of asthma[4]. Although inhaled glucocorticoids are effective at reducing morbidity and mortality, the global impact of asthma remains high, and the prevalence of the disease seems to be increasing

[1]Institute of Pediatrics, Children's Hospital of Fudan University, National Children's Medical Center, and the Shanghai Key Laboratory of Medical Epigenetics, International Co-laboratory of Medical Epigenetics and Metabolism, Ministry of Science and Technology, Institutes of Biomedical Sciences, Fudan University, Shanghai 200032, China. [2]National Health Commission (NHC) Key Laboratory of Neonatal Diseases, Fudan University, Shanghai 201102, China. [3]Department of Respiratory Medicine, Children's Hospital of Fudan University, National Children's Medical Center, Shanghai 201102, China. [4]Department of Critical Care Medicine, Children's Hospital of Fudan University, National Children's Medical Center, Shanghai 201102, China. [5]Department of General Medicine, Children's Hospital of Fudan University, National Children's Medical Center, Shanghai 201102, China. [6]Shanghai Children's Hospital, School of Medicine, Shanghai Jiao Tong University, Shanghai 200040, China. [7]These authors contributed equally: Xiao Han, Lijuan Liu, Saihua Huang, Wenfeng Xiao, Yajing Gao. ✉e-mail: sqhx12@126.com; llqian@126.com; yfzhou1@fudan.edu.cn

in developing countries[1,5]. A better understanding of the mechanisms underlying the pathogenesis of allergic asthma is critical for improving therapeutic outcomes.

Macrophages are the most abundant immune cells in the lung[6]. As important immune effector cells, macrophages display remarkable plasticity and change their physiology in response to environmental cues. Macrophage function or homeostasis is maintained by various regulatory processes, including the crosstalk between activation and autophagy[7]. The imbalance of macrophage homeostasis significantly affects immune-mediated lung injury associated with microbial infection, asthma, and tumorigenesis[8]. Macrophages are generally divided into classically activated (M1) and alternatively activated (M2) macrophages. The M1 subset, induced by IFN-γ/lipopolysaccharide (LPS), is involved in pro-inflammatory responses. In contrast, interleukin-4 (IL-4)/IL-13–induced M2 macrophages contribute to anti-inflammatory activity, wound healing, tissue repair, and allergic inflammation. A recently reported increased activation of M2 macrophages has been suggested to play a crucial role in allergic asthma through activating Th2 cell response[9–13]. However, how macrophage homeostasis is regulated in allergic asthma is less clear.

$N^6$-methyladenosine (m$^6$A) modification is the most prevalent internal modification in eukaryotic mRNAs, primarily affecting "RRACH" (R = G or A; H = A, C or U) consensus motif[14,15]. Methyltransferase-like 3 (METTL3), METTL14, and WTAP have been demonstrated to act as m$^6$A methyltransferases ('writers'), while modified m$^6$A is removed by demethylases, 'erasers', such as FTO and ALKBH5[16,17]. m$^6$A-binding proteins from the YT521-B homology (YTH) domain family, including YTHDF1/2/3, are known to be the 'readers' recognizing m$^6$A. Members of this family modulate the stability and translation of modified mRNAs[16,18,19]. It has been extensively documented that m$^6$A modification plays a crucial role in various biological and pathological processes, such as embryonic stem cell maintenance and differentiation[20], T-cell homeostasis[21], and tumorigenesis[22]. Accumulating evidence suggests that m$^6$A modifications influence the fate and function of mRNA, which may be critical in macrophage homeostasis[23,24]. Recently, *Mettl3*-deficient mice show increased M1/M2-like tumor-associated macrophage and regulatory T cell infiltration into tumors[25], while *Mettl3* deficiency in macrophages attenuates their ability to eliminate pathogens and tumors in vivo[26]. Nonetheless, the roles of m$^6$A modification and the underlying connection among m$^6$A 'writers', 'readers', and 'targets' in macrophage homeostasis and allergic asthma remain to be fully elucidated.

Here, we report a direct correlation between the decreased expression of *METTL3* in monocyte-derived macrophages and the severity of childhood allergic asthma. *Mettl3*-deficient myeloid cells promote Th2 cell response and aggravate airway inflammation in a mouse allergic asthma model by enhancing M2 macrophage activation. m$^6$A-seq and MeRIP-qPCR findings indicate that m$^6$A /METTL3/YTHDF3 interactions are involved in the repression of M2 macrophage activation by accelerating the degradation of Pentraxin 3 (*PTX3*) transcripts, which is widely regarded as a regulator of inflammation and asthma. Thus, our study illustrates that m$^6$A RNA modification regulates macrophage homeostasis and may provide therapeutic targets in the treatment of allergic asthma.

## Results

### *Mettl3* deficiency in myeloid cells aggravates airway inflammation in an allergic asthma model

m$^6$A methyltransferase, METTL3, has been reported to play a key role in the pathophysiology of several diseases by affecting diverse downstream genes[15,25,26]. To investigate the relevance of m$^6$A modifications in myeloid cell-mediated allergic asthma, we first crossed *Mettl3*$^{fl/fl}$ mice with *Lyz2*-Cre mice to ablate *Mettl3* in the myeloid compartment, including macrophages and neutrophils (Supplementary Fig. 1). Bone marrow-derived macrophages (BMDMs) from *Mettl3*$^{fl/fl}$ (WT) and

*Mettl3*$^{fl/fl}$*Lyz2*$^{Cre+}$ (*Mettl3* KO) mice were subjected to western blotting. BMDMs from the *Mettl3* KO mice showed a significant reduction in METTL3 protein levels, consistent with the reported ~70%-80% deletion efficiency of the *Lyz*-Cre system[27]. Compared to WT mice, no apparent abnormalities of external morphology were noted in *Mettl3* KO mice. *Mettl3* KO mice also developed normally and no spontaneous inflammatory pathologies were detected in various organs. We then sensitized WT and *Mettl3* KO mice with cockroach extract (CRE) to induce mouse allergic asthma model (Fig. 1a).

The results showed that, compared to PBS control, CRE-challenged mice exhibited a significant increase in the recruitment of inflammatory cells to the lungs, with dense peribronchial infiltrates, goblet cell hyperplasia, and mucus secretion. Compared to CRE-treated WT mice, the inflammatory response in CRE-treated *Mettl3* KO animals was particularly robust, with significantly increased numbers of total inflammatory cells in bronchoalveolar lavage fluid (BALF), including eosinophils (Fig. 1b, c). Furthermore, compared to WT mice, airway hyperresponsiveness (AHR) during the methacholine challenge was exaggerated in CRE-sensitized *Mettl3* KO animals (Fig. 1d). Histological examination confirmed the pronounced recruitment of inflammatory cells to the lung tissues in CRE-treated *Mettl3* KO mice, detecting dense peribronchial infiltrates, goblet cell hyperplasia, and mucus secretion (Fig. 1e, f), indicating that METTL3 may play a protective role during the pathogenesis of allergic asthma.

Given that neutrophils express high levels of *Lyz2* and can be recruited to lung tissues to exert pro-inflammation effects during asthma[2], we detected that the infiltration of Gr1$^+$ neutrophils in lung tissues tended to be enhanced in CRE-treated *Mettl3* KO mice compared with CRE-treated WT mice (Supplementary Fig. 2). To further examine whether the accelerated airway inflammation in *Mettl3* KO mice was related to the neutrophils, we depleted the neutrophils in CRE-induced asthma models by injecting anti-Ly6G Ab. Flow cytometry analysis and histological examination confirmed the depletion of neutrophils by anti-Ly6G Ab could not abrogate the increased airway inflammation phenotype of *Mettl3* KO mice (Supplementary Fig. 2). Thus, these findings indicated that the function of METTL3 in allergic airway inflammation was not dependent on neutrophils.

### Low *METTL3* expression in monocyte-derived macrophages from children with allergic asthma is associated with disease severity

To investigate the expression pattern of METTL3 in allergic asthma, we first analyzed microarray datasets from 5 human asthma patients and 5 healthy controls from the GEO database (GSE27876). The results showed that the expression of known m$^6$A writers (i.e., *METTL3* and *METTL14*), erasers (i.e., *FTO* and *ALKBH5*), and readers (i.e., *YTHDF3*) was moderately downregulated in peripheral blood mononuclear cells (PBMCs) of human asthma sufferers (Supplementary Fig. 3a). We next verified that, compared to 50 normal controls, the abundance of *METTL3*, *METTL14*, and *YTHDF3* was significantly decreased in PBMCs derived from 55 childhood allergic asthma patients, recruited from the Children's Hospital of Fudan University (Fig. 1g and Supplementary Fig. 3b). Similarly, *METTL3* expression was lower in primary monocyte-derived macrophages of childhood allergic asthma patients than in normal controls (Fig. 1h and Supplementary Fig. 3c).

These observations led us to hypothesize that METTL3 may be associated with the severity of childhood allergic asthma. Thus, the ability of monocyte-derived macrophages *METTL3* abundance to differentiate childhood asthma patients from healthy participants was firstly assessed by receiver operating characteristic (ROC) curve analysis, yielding area under the curve (AUC) values of 0.79 (Fig. 1i). Moreover, we explored the correlation between monocyte-derived macrophage *METTL3* levels and disease severity among patients with asthma. A negative correlation was observed between *METTL3* expression and blood eosinophil numbers, and fraction of exhaled

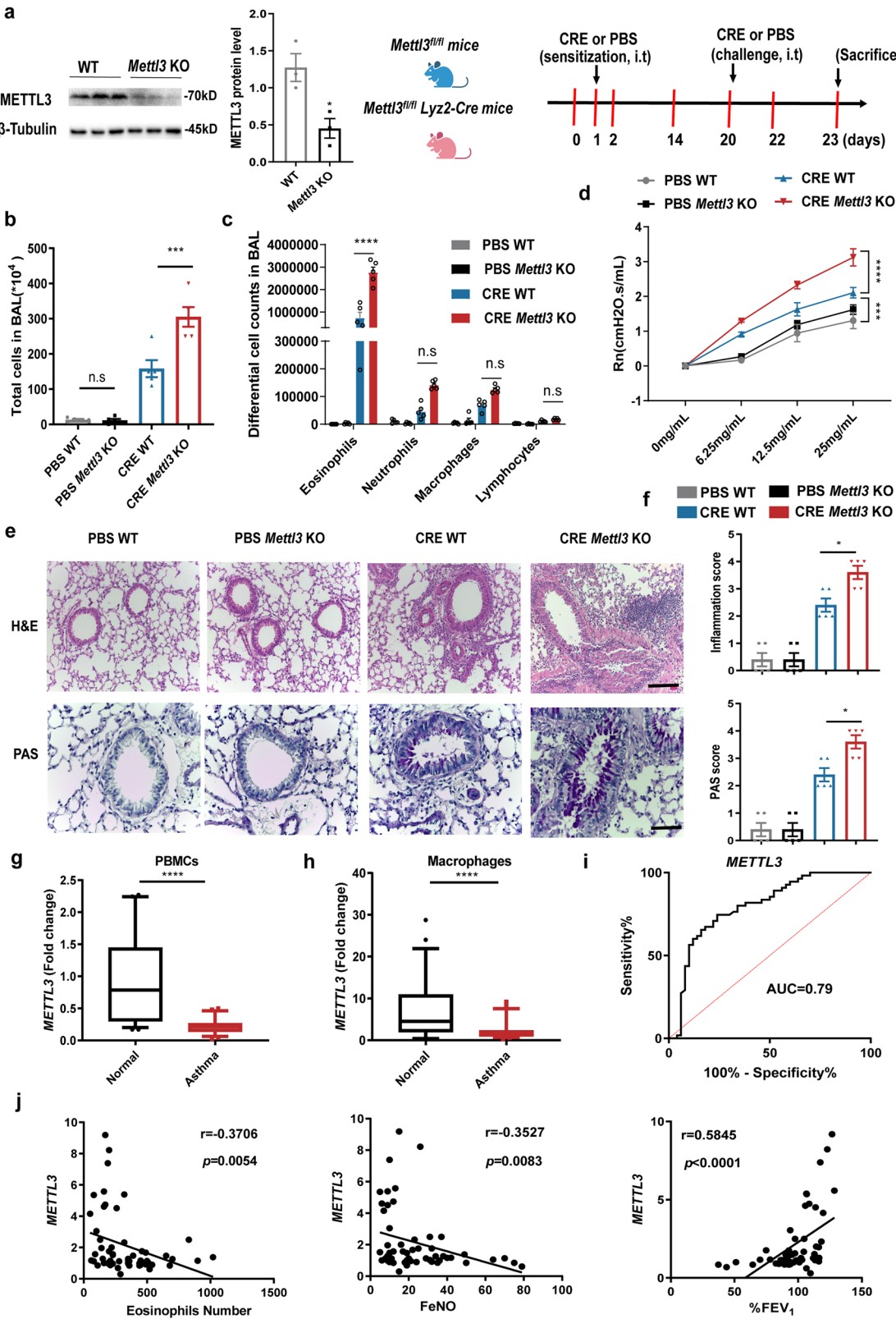

nitric oxide (FeNO) levels, confirming the known eosinophilic airway inflammation in allergic asthma patients. In contrast, low levels of *METTL3* were associated with the worse pulmonary function (percent predicted forced expiratory volume in 1 s, %FEV$_1$) in patients (Fig. 1j). Together, these data suggest a possible association between the severity of childhood allergic asthma and a decreased *METTL3* expression in monocyte-derived macrophages.

## METTL3-deficient macrophages are more susceptible to M2 activation through the PI3K/AKT and JAK/STAT6 signaling, thereby accelerating allergic airway inflammation

To further confirm whether the role of METTL3 in allergic airway inflammation is due to macrophage-intrinsic effect, we depleted macrophages in vivo using clodronate-containing liposomes (CLs)[28,29]. Flow cytometry analysis confirmed a significant decrease in the

**Fig. 1 | Mettl3 deficiency in myeloid cells aggravates airway inflammation in allergic asthma. a** Schematic illustration of the cockroach extract (CRE)-induced mouse allergic asthma model (Schematic created with BioRender.com). METTL3 protein levels were determined in BMDMs from WT (*Mettl3*fl/fl) and *Mettl3* KO (*Mettl3*fl/fl*Lyz2*Cre+) mice by Western blot (*n* = 3 animals). **b** Total and (**c**) differential BALF cell numbers in PBS- and CRE-treated WT and *Mettl3* KO mice as detected by flow cytometry (*n* = 5 animals). **d** Assessment of methacholine-induced airway hyperresponsiveness (AHR) in mice (*n* = 4 animals for CRE WT group and *n* = 5 animals for other groups). Central airway resistance (Newtonian resistance, Rn) values are shown. **e** Representative images of H&E- and PAS-stained lung tissues (scale bars: 200 μm and 100 μm, respectively). **f** Inflammation and PAS scores were quantified (*n* = 5 animals). RT-qPCR analysis of the transcripts levels of *METTL3* in PBMCs (**g**) and monocyte-derived macrophages (**h**) from 55 childhood allergic asthma and 50 healthy controls, respectively. The detailed minimum, median, maximum, 25th, 75th percentile (box), and 5th and 95th percentile (whiskers) of box plots were provided in the Source data file. **i** Receiver operating characteristic (ROC) curve analysis of monocyte-derived macrophages *METTL3* levels in the 55 childhood asthma population. **j** Spearman correlation (two-sided) analysis of monocyte-derived macrophages *METTL3* expression, blood eosinophils number, FeNO, and %FEV1 levels in 55 childhood asthma. Statistical analysis of the data was performed using two-sided unpaired *t* test (**a**), Mann–Whitney test (**g**, **h**), 1-way ANOVA (**b**, **f**), and 2-way ANOVA (**c**, **d**) followed by Tukey's multiple comparison tests. Data are presented as means ± SEM from one of three independent experiments. *$P < 0.05$, ***$P < 0.001$, ****$P < 0.0001$; n.s, not significant.

percentage of alveolar macrophages in BALF from *Mettl3* KO mice treated with CLs, compared to controls (PBS)-treated *Mettl3* KO mice (Fig. 2a). In addition, the depletion of macrophages completely reversed the susceptibility of *Mettl3* KO mice to allergic airway inflammation, compared with wild-type animals (Fig. 2b–d), suggesting that the vital role of METTL3 in airway inflammation is dependent on macrophages.

Next, to characterize the potential involvement of METTL3 in macrophage function, transcriptome sequencing (RNA-seq) was performed in human *METTL3*-deficient THP1-derived macrophages, following IL-4 stimulation. We identified 266 genes that were differentially expressed in *METTL3*-depleted macrophages (≥1.5-fold change, $P < 0.05$). Of these 44% (118 genes) were downregulated and 56% (148 genes) were upregulated. Notably, genes associated with M2 macrophage activation, including *IL-10*, *CD206*, *PPARG*, and *KLF4*, were robustly upregulated in *METTL3*-depleted macrophages (Fig. 2e). To validate this RNA-seq data, BMDMs were stimulated with LPS to induce M1 differentiation or with IL-4 to induce the M2 subset. RT-qPCR results showed that compared to WT mice, the expression of M2 macrophage activation-associated markers (i.e., *Il-10*, *Cd206*, and *Arg1*) was significantly increased in BMDMs from *Mettl3* KO mice, while M1-associated markers (i.e., *Tnf, Il-6*, and *Il-1β*) were suppressed (Fig. 2f). ELISA experiments revealed that IL-10 was also upregulated at the protein level, while TNF was comparatively reduced (Fig. 2g). Conversely, gain functional studies in BMDMs with *METTL3* overexpression (*METTL3* LV) demonstrated higher levels of M1-associated, and lower levels of M2-associated markers (Supplementary Fig. 4). Consistently, we also found that *METTL3* in human THP1-derived macrophages had the same regulatory effect on the M1/M2 activation (Supplementary Fig. 5a-c). Therefore, these findings confirmed that METTL3 deficiency could induce a bias towards M2 type during macrophage activation while inhibiting M1 activation.

In addition, the Kyoto Encyclopedia of Genes and Genomes (KEGG) pathway gene set data was analyzed in *METTL3*-deficient macrophages. RNA-seq data indicated that a small number of differentially expressed genes were associated with Phosphatidylinositol 3-kinase-AKT (PI3K/AKT) signaling, Janus kinase-signal transducer, and activator of transcription (JAK/STAT) signaling, and the Autophagy pathway (Fig. 2h). It is well-known that the PI3K/AKT and JAK/STAT6-signaling pathways play a crucial role in promoting M2 macrophage activation[30,31]. To further explore whether the PI3K/AKT and JAK/STAT6 signaling pathways were necessary for METTL3-mediated M2 activation, we performed immunoblots, analyzing BMDMs from WT and *Mettl3* KO mice. As expected, upon IL-4 stimulation, the phosphorylation levels of AKT and STAT6 in *Mettl3*-deficient BMDMs were markedly increased. However, the total levels of these proteins were comparable, irrespective of whether BMDMs were derived from *Mettl3*-deficient or WT animals (Fig. 2i). A similar enhanced phosphorylation of AKT and STAT6 proteins was noted in human THP1-derived macrophages after *METTL3* knockdown, whereas their phosphorylation were reduced in *METTL3*-overexpressing cells (Supplementary Fig. 5d). Furthermore, the rescue experiments demonstrated

that *Mettl3* deficiency enhanced M2-associated genes levels, whereas the inhibition of AKT or STAT6 phosphorylation levels eliminated this effect (Supplementary Fig. 5e). Notably, in accordance with a previously published study[26], we also found that *METTL3* deficiency inhibited M1 macrophage activation through negatively regulating NF-κB signaling (Supplementary Fig. 6). Taken together, these data indicate a vital role for METTL3 in balancing the equilibrium between M1/M2 macrophage activation by repressing M2 activation via the PI3K/AKT and JAK/STAT6 signaling.

## *Mettl3* depletion enhances Th2 cell response in allergic airway inflammation by facilitating M2 macrophage activation

Previous work has demonstrated the importance of increased M2 macrophage activation in the progression of allergic asthma[11,12]. To further investigate whether *Mettl3* depletion exacerbated allergic airway inflammation through modulating M2 macrophage activation, we purified alveolar macrophages from lung tissues. The mRNAs and protein levels of M2-associated genes were significantly elevated in alveolar macrophages from CRE-challenged *Mettl3* KO mice, compared to CRE-challenged WT animals (Fig. 3a, b). Furthermore, flow cytometry analysis demonstrated that alveolar macrophages from CRE-challenged *Mettl3* KO mice expressed more CD206 than the same cells from CRE-challenged WT mice (Fig. 3c, d). In accordance with the flow cytometry data, immunofluorescence staining also revealed that the number of M2 macrophages (F4/80+CD206+) increased significantly in the lung tissues of CRE-treated *Mettl3* KO mice (Fig. 3e). Thus, these data demonstrated that *METTL3* deficiency significantly promoted M2 macrophage activation during allergic airway inflammation.

Allergic asthma has been generally considered a Th2 cell-mediated chronic immune response, while Th2 cells produce effector cytokines such as IL-4 and IL-13 to mediate respiratory symptoms. More importantly, accumulating evidence reveals that in allergic asthma, M2 macrophages release high levels of chemokines, including CCL-17 and CCL-22, resulting in the recruitment of Th2 cells and amplification of polarized Th2 cell responses in the bronchial tissue[9,32]. To further determine whether *METTL3* depletion promotes Th2 responses in allergic inflammation via M2 macrophage activation, the levels of M2-associated chemokines (i.e., *Ccl-17*, and *Ccl-22*) in alveolar macrophages were tested. We found that the mRNA levels of *Ccl-17* and *Ccl-22* were significantly elevated in alveolar macrophages from CRE-challenged *Mettl3* KO mice (Fig. 3f). Furthermore, compared to WT mice, the mRNA and protein levels of Th2 cell-associated cytokines (i.e., *IL-4*, and *IL-13*) were markedly increased in lung homogenates from CRE-challenged *Mettl3* KO mice, whereas the Th1 cell-associated cytokine (*ifng*) showed downregulated expression and the Th17 cell-associated cytokine (*Il-17a*) had no difference (Fig. 3g, h). Lastly, the data demonstrated that IL-4-producing Th2 cells were also enhanced in the mediastinal lymph nodes (MLNs) from CRE-challenged *Mettl3* KO mice, while IFN-gamma−producing Th1 cells were comparatively reduced (Fig. 3i, j). Collectively, the above findings suggest that *METTL3* deficiency promotes Th2 responses and accelerates allergic airway inflammation via enhancing M2 macrophage activation.

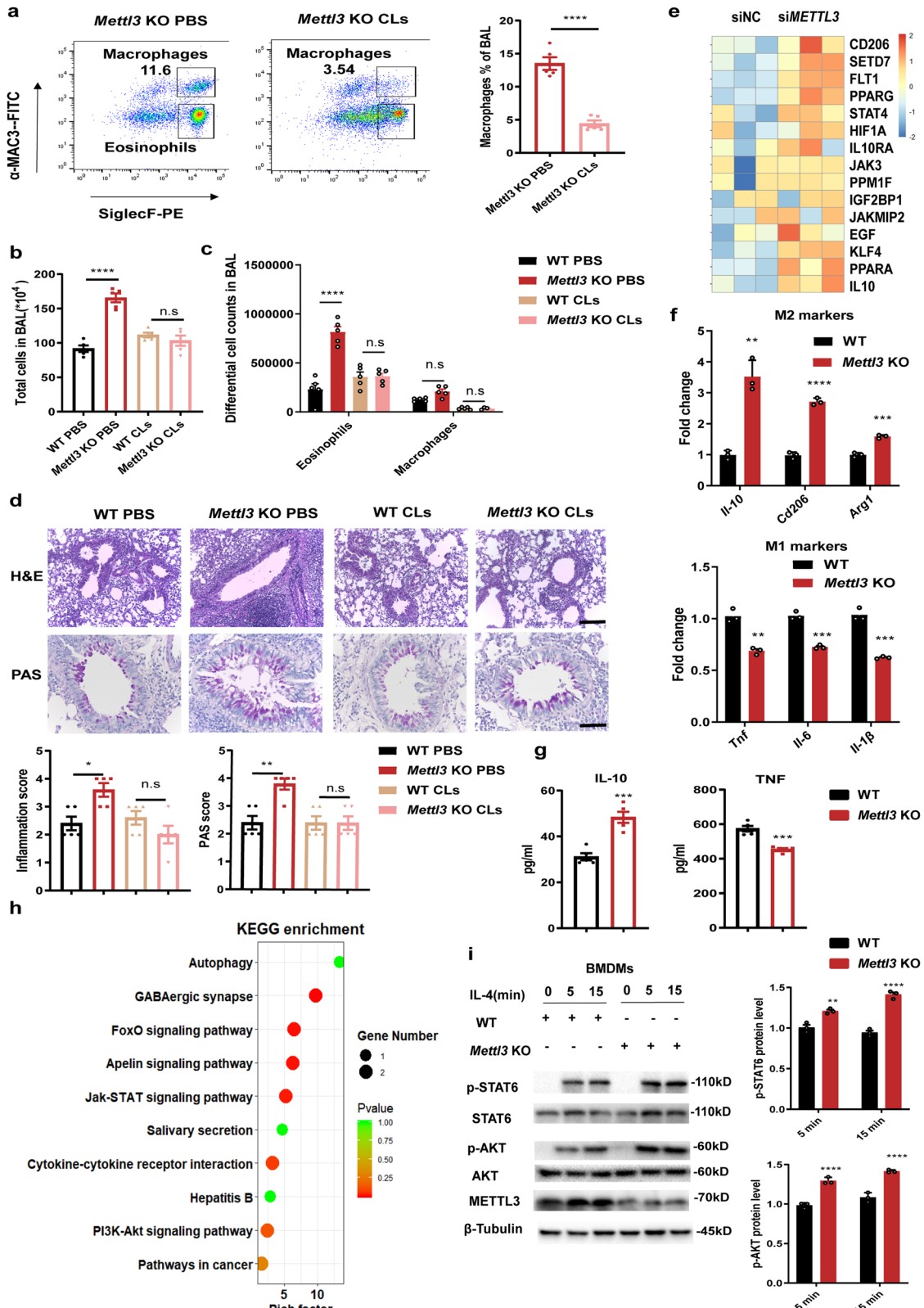

## PTX3 is a direct target of METTL3

To identify critical downstream targets of METTL3-mediated m6A modification in macrophages, we mapped m6A methylomes of *METTL3*-deficient and control THP1-derived macrophages by m6A sequencing (m6A-seq) and RNA-seq. Of the 724 m6A peaks showing pronounced changes after *METTL3*-depletion, the vast majority (668, 93.3%) exhibited a significant decrease in m6A abundance

(Supplementary Fig. 7a). The previously reported consensus "RRACH" motif (R = G or A; H = A, C or U) was identified in both control and *METTL3*-deficient cells (Fig. 4a). In addition, we detected that the density of m6A was markedly enriched near stop codons (Fig. 4b), consistent with previous observations[15,33]. Meanwhile, a similar pattern of total and common m6A distribution was seen in control and *METTL3*-deficient macrophages, with m6A peaks being particularly abundant in

**Fig. 2 | METTL3-deficient macrophages are more susceptible to M2 activation through the PI3K/AKT and JAK/STAT6 signaling, thereby accelerating allergic airway inflammation. a** CLs-liposome was i.p. administered in the clodronate group and PBS was administered in the control group. Flow cytometry analysis of the efficiency of macrophage depletion in BALF from *Mettl3* KO mice by clodronate treatment (*n* = 5 animals). **b** Total and (**c**) differential BALF cell numbers from experimental asthmatic animals were analyzed by flow cytometry (*n* = 5 animals). **d** Histopathological changes in the lung tissues were examined by H&E- and PAS-staining. Scale bars: 200 μm and 100 μm, respectively. Inflammation and PAS scores were quantified (low panel) (*n* = 5 animals). **e** Heatmap illustrating the upregulated expression of M2 activation-associated genes in human THP1-derived macrophages transfected with *METTL3* siRNA pools (200 nM), compared with negative control (NC) cells. **f** RT-qPCR analysis of M1 and M2-associated markers expression in WT and *Mettl3* KO mice BMDMs (*n* = 3 animals) stimulated with LPS or IL-4, respectively. **g** ELISA shows the levels of TNF and IL-10 secretion in WT and *Mettl3* KO mice BMDMs (*n* = 5 animals). **h** KEGG enrichment analysis shows the top pathways enriched in *METTL3*-deficient macrophages. **i** Western blot showing levels of AKT phosphorylation (p-AKT), AKT, STAT6, and p-STAT6 in WT and *Mettl3* KO BMDMs (*n* = 3 animals), following IL-4 stimulation. Statistical analysis of the data was performed using two-sided unpaired *t* test (**a**, **f**, **g**, **h**), 1-way ANOVA (**b**, **d**), and 2-way ANOVA (**c**, **i**) followed by either Tukey's or Sidak's multiple comparison tests. Data are presented as means ± SEM from one of three independent experiments. *\*P* < 0.05, *\*\*P* < 0.01, *\*\*\*P* < 0.001, *\*\*\*\*P* < 0.0001; n.s, not significant.

the vicinity of CDS and 3′UTR regions (Fig. 4c). We then analyzed the frequency and distribution of m⁶A modifications within differentially expressed mRNAs, selecting m⁶A peaks and mRNA abundance with > 1.2-fold change, *P* < 0.05. There were 152 hypo-methylated m⁶A peaks whose mRNA transcripts were either significantly downregulated (88, Hypo-down) or upregulated (64, Hypo-up) in *METTL3*-deficient macrophages. In contrast with the high number of hypo-methylated m⁶A peaks, only 26 hyper-methylated m⁶A peaks were detected. Half of these were downregulated (13 Hyper-down), while the other half showed upregulation (13 Hyper-up) (Fig. 4d). Considering the role of METTL3 in the m⁶A 'writers' complex, up-regulated mRNAs transcripts carrying hypo-methylated m⁶A peaks in *METTL3*-deficient macrophages were the likely potential downstream targets of METTL3.

Next, we characterize potential targets involved in METTL3-regulated M2 activation. The 64 m⁶A-Hypo-up genes were intersected with 28 M2 activation-associated genes with known regulatory functions (listed in Supplementary Table 1). This comparison identified only three overlapping genes, namely pentraxin-3 (*PTX3*), C-C Motif Chemokine Ligand 13 (*CCL13*), and RIP-Like Protein Kinase 3 (*RIPK3*) (Fig. 4e). Among these three, PTX3 has been recently reported to be a positive regulator of M2 macrophage activation in the development of inflammatory diseases[34,35]. Interestingly, we noted that *PTX3* levels exhibited the inverse correlation with *METTL3* in monocyte-derived macrophages from childhood allergic asthma patients (Fig. 4f). Meanwhile, the m⁶A-seq data identified that the *PTX3* m⁶A level was markedly enriched in control cells compared to the *METTL3*-deficient cells (Fig. 4g), while *PTX3* harbored two m⁶A sites in its 3′ UTR (AGA$^{1333}$CT and AGA$^{1406}$CA, respectively). Subsequently, the SELECT analysis demonstrated that the relative amount of qPCR products targeting the AGA$^{1333}$CT site was significantly enhanced in the *METTL3*-deficient macrophages compared to the control, whereas no significant difference was observed in the amplification of the AGA$^{1406}$CA site, implying the AGA$^{1333}$CT site could be methylated by METTL3 (Fig. 4h and Supplementary Fig. 7b). Furthermore, based on the AGA$^{1333}$CT site in *PTX3* transcripts, we performed MeRIP assays and confirmed that the enrichment m⁶A level of *PTX3* transcripts was significantly downregulated in *METTL3*-deficient macrophages (Fig. 4i). In addition, *METTL3* deficiency in both mouse BMDMs and human THP1-derived macrophages led to noticeably elevated *PTX3* mRNA and protein levels after IL-4 stimulation, whereas overexpression of *METTL3* resulted in a remarkable downregulation (Fig. 4j, k, and Supplementary Fig. 8), indicating that *PTX3* is a target of METTL3.

To further confirm that the regulation of *PTX3* by METTL3 relies on the methylation of mRNA transcripts, we performed luciferase reporter assays with the *PTX3* 3′UTR constructs containing wild-type or mutant m⁶A sites (m⁶A modified by A$^{1333}$ to T substitution). *METTL3* depletion in THP1-derived macrophages significantly enhanced the luciferase activity of the reporter construct carrying the *PTX3* 3′UTR. This increase was abrogated when the putative m⁶A sites were mutated (Fig. 4l), suggesting that the downregulation of *PTX3* expression by METTL3 was dependent on m⁶A modification of the 3′UTR within *PTX3*.

It is well-known that m⁶A modification of mRNA transcripts affects mRNA stability[18,22,36]. To investigate whether METTL3 regulated *PTX3* expression by this mechanism, we treated WT or *Mettl3* KO BMDMs with the transcription inhibitor actinomycin D and followed the decay of *PTX3* transcripts. *Mettl3* depletion in these experiments markedly increased *PTX3* mRNA stability (Fig. 4m). Collectively, these findings imply that METTL3 enhances the m⁶A modification of the 3′UTR of *PTX3*, leading to the instability and degradation of *PTX3* transcripts.

## Disruption of *PTX3* attenuates METTL3 depletion-induced M2 macrophage activation and allergic airway inflammation

Next, we explored whether the effect of METTL3 on M2 macrophage activation was mediated via PTX3. The levels of M2-associated markers were significantly downregulated in *Ptx3* knockdown BMDMs. This coincided with a notable reduction of PI3K/AKT and JAK/STAT6 signaling in IL-4-treated M2 macrophages (Fig. 5a, b). Similar observations were made in THP1-derived macrophages after *PTX3* knockdown (Supplementary Fig. 9), implying that the downregulation of *PTX3* inhibited M2 macrophage activation. Additionally, the data detected that the enhanced effect of *Mettl3* deficiency on M2 macrophage activation was markedly abolished by *Ptx3* knockdown (Fig. 5c). These results suggest that the upregulation of *PTX3* is responsible for the preferential M2 macrophage activation seen in *METTL3*-deficient macrophages.

Thus, we hypothesized that the overexpression of *PTX3* may be responsible for the enhanced allergic inflammation in *Mettl3* deficient mice. To test this, we first depleted macrophages in vivo using clodronate-containing liposomes (CLs). In contrast with our findings in *Mettl3* KO mice, the data detected that compared to shCtrl-infected mice, *Ptx3* knockdown in vivo noticeably alleviated the CRE-induced allergic airway inflammation. However, we found that, through the depletion of macrophages, there was no significant difference in airway inflammation between sh*Ptx3*–treated mice (sh*Ptx3* CLs) and shCtrl-mice (shCtrl CLs) (Supplementary Fig. 10), suggesting that the role of *Ptx3* knockdown in allergic airway inflammation may be due to macrophages. Subsequently, rescue experiments in vivo were performed by injecting sh*Ptx3* or control (shCtrl) lentiviruses into CRE-sensitized mice. As expected, the increased concentration of serum PTX3 in CRE-treated *Mettl3* KO mice was largely reversed by the shRNA-mediated knockdown of *Ptx3* (Fig. 5d). In addition, BALF from sh*Ptx3*-infected *Mettl3* KO mice contained a remarkably lower number of total inflammatory cells than those from shCtrl-infected *Mettl3* KO animals. A prominent reduction in the number of eosinophils was also seen in sh*Ptx3*-infected animals (Fig. 5e, f). AHR, peribronchial inflammation, goblet cell hyperplasia, and mucus secretion seen in *Mettl3* KO mice were substantially diminished after infection with the sh*Ptx3* lentiviruses (Fig. 5g–i). Overall, these data confirmed that reducing *Ptx3* could attenuate airway inflammatory responses in *Mettl3* deficiency animals.

Importantly, we also found that by inhibiting the expression of *Ptx3*, the abundance of M2 activation-associated markers could be substantially reduced in alveolar macrophages (Fig. 5j, k). Moreover,

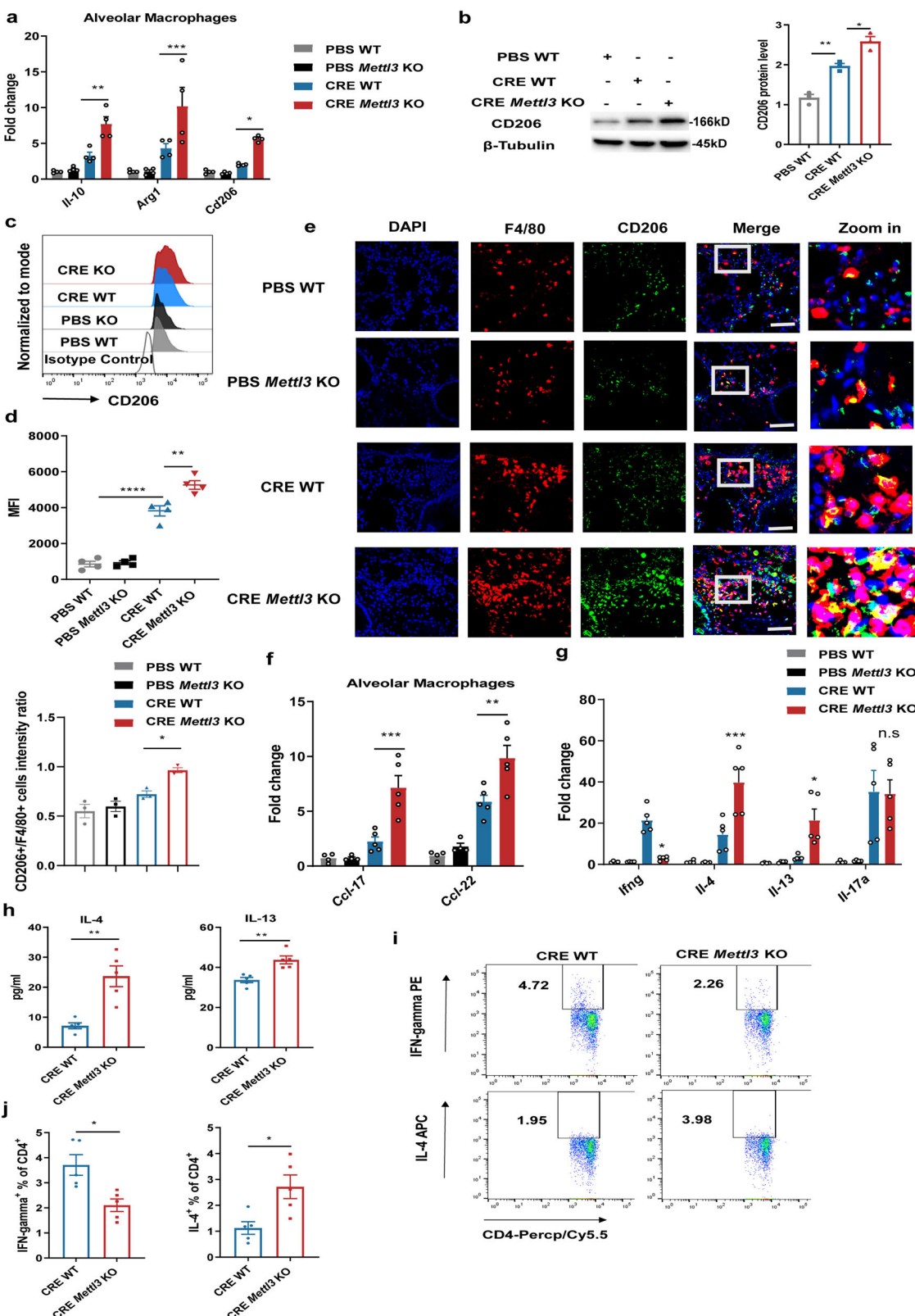

flow cytometry analysis showed that alveolar macrophages from sh*Ptx3*-infected *Mettl3* KO mice expressed lower CD206 levels than those from shCtrl-infected *Mettl3* KO mice (Fig. 5l). Immunofluorescence staining detected a similar decline in the number of M2 macrophages (F4/80⁺ CD206⁺) in the lung tissues from sh*Ptx3*-treated *Mettl3* KO mice, compared to shCtrl-treated *Mettl3* KO animals (Fig. 5m). Taken together, the above data strongly suggest that

METTL3 inhibits M2 macrophage activation and alleviates airway inflammation in allergic asthma via repressing *PTX3* expression.

### Loss of *METTL3* impairs the YTHDF3-mediated degradation of *PTX3* mRNA

Recent studies have illustrated the importance of the family of YTHDF proteins, m⁶A 'readers', in the regulation of physiological processes

**Fig. 3 | *Mettl3* depletion enhances Th2 cell response in allergic airway inflammation via facilitating M2 macrophage activation. a** The mRNA expression of M2-associated markers (*n* = 4 animals), and (**b**) CD206 protein levels (*n* = 3 animals) in alveolar macrophages purified from CRE-induced asthma models were determined, using RT-qPCR and Western blot, respectively. **c** Flow cytometry analysis of M2 macrophage subpopulation in BALF from experimental models. **d** The MFI of M2 macrophages was calculated (*n* = 4 animals). **e** Representative composite fluorescent images showing the number of M2 macrophages (F4/80⁺ CD206⁺) in lung tissues from experimental models. Total macrophages, M2 macrophages, and nuclei were labeled with F4/80 (red), CD206 (green), and DAPI (blue), respectively. Scale bars: 25 μm. Quantification of the CD206 intensity ratio in macrophages was performed by Image J (bottom) (*n* = 3 animals). **f** RT-qPCR showing up-regulated M2-associated chemokines in alveolar macrophages purified from experimental models (*n* = 4 animals for PBS groups and *n* = 5 animals for CRE groups). The levels of Th1, Th2, and Th17 cell-associated cytokines in lung homogenates were detected by RT-qPCR (**g**) (*n* = 4 animals for PBS groups and *n* = 5 animals for CRE groups) and ELISA (**h**) (*n* = 5 animals), respectively. **i, j** Flow cytometry analysis of the frequency of CD4⁺ IL-4⁺ Th2 cells and CD4⁺ IFN-gamma⁺ Th1 cells in MLNs from mice (*n* = 5 animals). Statistical analysis of the data was performed using two-sided unpaired *t* test with Welch's correction (**h** left) or not (**a**, **h** right, **j**), 1-way ANOVA (**b**, **d**, **e**), and 2-way ANOVA (**f**, **g**) followed by Tukey's multiple comparison tests. Data are presented as means ± SEM from one of three independent experiments. \**P* < 0.05, \*\**P* < 0.01, \*\*\**P* < 0.001, \*\*\*\**P* < 0.0001; n.s, not significant.

and various diseases[16,18]. Interestingly, during the analysis of down-regulated genes in *METTL3*-deficient macrophages, we noted that the mRNA level of *YTHDF3* was robustly decreased in *METTL3*-deficient macrophages (Fig. 6a). Consistent with the findings in RNA-seq, YTHDF3 protein levels were suppressed in both *Mettl3* KO BMDMs and *METTL3*-deficient THP1-derived macrophages (Fig. 6b). It was notable that *YTHDF3* abundance was also significantly reduced in the monocyte-derived macrophages from childhood allergic asthma (Fig. 6c). YTHDF3 has been shown to play a key role in mediating the degradation and translation of m⁶A-modified mRNAs[19,37,38]. To further verify whether YTHDF3 was involved in the effect of METTL3 on the M2 macrophage activation, we knocked down the expression of *YTHDF3* in human THP1-derived macrophages. The results revealed that the loss of *YTHDF3* enhanced the expression of M2 activation-associated genes, accompanied by increased phosphorylation of STAT6 and AKT (Fig. 6d, e). These findings were consistent with observations made in *METTL3*-deficient macrophages. In addition, rescue experiments demonstrated that enforced expression of *METTL3* inhibited M2-associated genes levels, whereas knockdown of *YTHDF3* abolished this effect (Fig. 6f). Therefore, the regulatory function of METTL3 on M2 macrophage activation seems dependent on the presence of YTHDF3.

Moreover, to confirm whether METTL3 represses *PTX3* expression through YTHDF3-mediated m⁶A modification, we performed RIP-qPCR assays targeting the m⁶A site in *PTX3* transcripts. As expected, *PTX3* m⁶A transcripts were remarkably enriched in YTHDF3-pull precipitates, while this relative enrichment was significantly decreased in *METTL3*-deficient macrophages (Fig. 6g). Furthermore, similar to our findings in *METTL3*-depleted macrophages, elevated *PTX3* mRNA and protein levels were also found in human *YTHDF3* knockdown macrophages (Fig. 6h, i). Rescue experiments revealed that the decreased *PTX3* levels seen in *METTL3*-overexpressing human macrophages could be largely reversed by the knockdown of *YTHDF3* (Supplementary Fig. 11). mRNA decay assays demonstrated that the degradation of *PTX3* transcripts was appreciably attenuated in *YTHDF3*-deficient macrophages (Fig. 6j). In addition, luciferase reporter assays showed that *YTHDF3* knockdown significantly increased the activity of a luciferase construct containing the *PTX3* 3′UTR, but such increment was largely impaired in *PTX3* mutant 3′UTR (Fig. 6k), confirming the m⁶A-dependent regulation of *PTX3* expression. Overall, these findings indicate that the METTL3/YTHDF3 m⁶A axis directly downregulates *PTX3* expression in macrophages by reducing its mRNA stability.

## METTL3/YTHDF3-m⁶A/PTX3/STX17 axis controls autophagy maturation in macrophages

To further explore the roles of PTX3 in macrophage functions, we performed RNA-seq in *PTX3* knockdown macrophages stimulated with IL-4. As expected, KEGG pathway analysis revealed that the differently expressed transcripts were enriched in PI3K/AKT and JAK/STAT signaling pathways. Interestingly, genes involved in the autophagy pathway were also enriched in *PTX3*-deficient macrophages, consistent with previous findings in *METTL3*-depleted cells. Specifically, the data showed that genes enriched in autophagy were associated with the

soluble N-ethylmaleimide-sensitive factor attachment protein receptor (SNARE) complexes (Fig. 7a), which are essential for the fusion between autophagosome and lysosome (also called autophagy maturation), and autophagosome degradation[39,40]. Since autophagy is crucial to promote M2 macrophage activation[41,42], we speculated that PTX3 was required for autophagy-mediated macrophage homeostasis. To test this, we performed transmission electron microscopy. The data showed that stimulating with rapamycin and IL-4, *Ptx3* knockdown in macrophages exhibited the up-regulated number of autophagosomes, suggesting either an increased autophagy flux or a defect of autophagosome degradation (Fig. 7b). We then transduced an adenovirus plasmid encoding mRFP-GFP-LC3 into control and *Ptx3*-deficient BMDMs. After inducing autophagy, the GFP was degraded in control cells but the majority of LC3 dots in *Ptx3*-deficient cells remained RFP and GFP double-positive, indicating the accumulation of autophagosomes without degradation in *Ptx3*-deficient cells (Fig. 7c). In contrast, the number of autophagosomes was significantly reduced in both *Mettl3* KO BMDMs and *YTHDF3* knockdown THP1-derived macrophages (Fig. 7d, e). Additionally, as compared with the control cells, the majority of LC3 dots in both *Mettl3* KO BMDMs and *YTHDF3* knockdown THP1-derived macrophages just remained RFP-positive (Fig. 7f, g), implying the marked degradation of autophagosomes in these cells. Collectively, these findings imply that the METTL3/YTHDF3-m⁶A/PTX3 axis in macrophages exerts regulatory effects on autophagy maturation.

More importantly, the RNA-seq analysis identified that syntaxin 17 (*STX17*), a vital component of SNARE complexes, was among the top down-regulated genes in *PTX3* deficient macrophages (Fig. 7h). *PTX3* knockdown in both BMDMs and THP1-derived macrophages strongly decreased *STX17* mRNA and protein expression (Fig. 7i and Supplementary Fig. 12a). By contrast, the STX17 levels were enhanced in both *Mettl3* KO BMDMs and *YTHDF3* knockdown THP1-derived macrophages (Fig. 7j, k), indicating that METTL3/YTHDF3-m⁶A/PTX3 axis regulates the STX17 expression. Next, to explore whether the PTX3/STX17 axis regulated autophagy maturation in macrophages, *Ptx3* knockdown and control BMDMs were treated with rapamycin and the level of LC3-II was measured. Expectedly, STX17 expression was significantly reduced in *Ptx3*-deficient BMDMs, although, with time, LC3-II was eventually degraded in control but not *Ptx3*-deficient cells (Fig. 7l). Similar findings were obtained in *PTX3*-deficient THP1-derived macrophages (Supplementary Fig. 12b), implying a defect of autophagy maturation-mediated autophagosome degradation in *Ptx3*-deficient macrophages. Notably, our rescue experiments demonstrated that the downregulated autophagosomes seen in *Mettl3* KO BMDMs could be largely reversed by the knockdown of *Stx17* (Supplementary Fig. 12c). Thus, the data above suggest that the regulatory effect of the METTL3/YTHDF3-m⁶A/PTX3 axis on macrophage homeostasis may be partially related to its role in autophagy maturation via an STX17-dependent manner.

## PTX3 expression is enhanced in both monocyte-derived macrophages and plasma from children with allergic asthma

PTX3 has been studied as an emerging biomarker reflecting tissue injuries and inflammation, such as lung carcinoma and heart

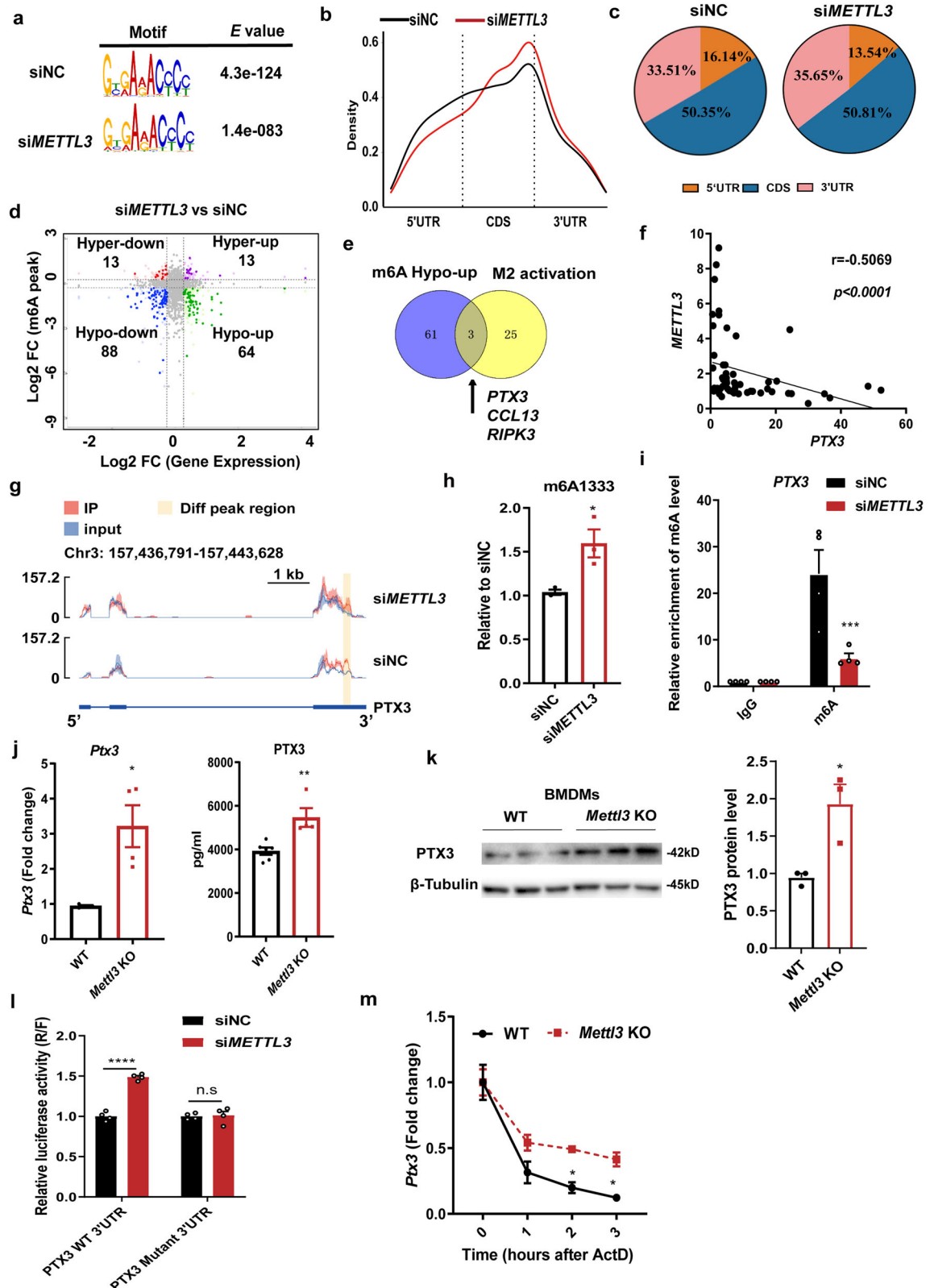

failure[43,44]. To further elucidate the potential importance of PTX3 in allergic asthma, we evaluated its expression in childhood allergic asthma and healthy controls. In line with observations in the mouse model, both the *PTX3* mRNA levels in monocyte-derived macrophages and PTX3 circulating levels in plasma were higher in childhood allergic asthma than in healthy participants (Fig. 8a, b). Importantly, *STX17*, as a key target of PTX3-mediated macrophage

homeostasis, was also more abundant in monocyte-derived macrophages isolated from asthma patients (Fig. 8c). ROC curve analysis, yielding AUC values of 0.69, indicated that PTX3 circulating levels in plasma may be useful for diagnosing childhood allergic asthma (Fig. 8d). Additionally, PTX3 circulating levels showed a positive correlation with blood eosinophils numbers and FeNO levels in childhood allergic asthma (Fig. 8e). Taken together, these findings

**Fig. 4 | METTL3 regulates *PTX3* mRNA stability and expression through manipulation of m⁶A modification. a** The predominant consensus m⁶A motif RRACH was detected in both control and *METTL3*-deficient macrophages through m⁶A-seq. **b** Density distribution of m⁶A peaks across mRNA transcripts. **c** The proportion of m⁶A peaks distribution in the 5′UTR, CDS, and 3′UTR regions across mRNA transcripts in control and *METTL3*-deficient macrophages. **d** Distribution of peaks (>1.2-fold change, $P < 0.05$) with a significant change in both mRNAs expression and m⁶A level in *METTL3*-deficient macrophages relative to control cells. **e** Venn diagram showing the number of overlapping genes between m⁶A-Hypo-upregulated genes in *METTL3*-deficient macrophages and M2 activation-associated functional genes. **f** Spearman correlation (two-sided) between *METTL3* and *PTX3* mRNA levels in monocyte-derived macrophages from 55 childhood allergic asthma populations. **g** Integrative Genomics Viewer (IGV) showing the m⁶A abundance on *PTX3* mRNA transcripts in *METTL3* deficiency and control macrophages as detected by m⁶A-seq. **h** SELECT assay showing the relative amount of qPCR products targeting the AGA$^{1333}$CT site on *PTX3* 3′UTR in THP1-derived macrophages transfected with *METTL3* or control siRNA ($n = 3$ cells). **i** MeRIP-qPCR assay validated the enrichment of m⁶A-modified *PTX3* transcripts in *METTL3*-deficient THP1-derived macrophages ($n = 4$ cells). **j** RT-qPCR ($n = 4$ animals) and ELISA ($n = 6$ animals for WT group and $n = 4$ animals for KO group) showing increased *PTX3* mRNA and secretion levels in *Mettl3* KO BMDMs. **k** Protein levels of PTX3 in WT and *Mettl3* KO BMDMs were detected by Western blot ($n = 3$ animals). **l** Luciferase reporter and mutagenesis assays. Wild-type (WT) or mutant (m⁶A$^{1333}$ was replaced by T) *PTX3*-3′UTR vector-transfected THP1-derived macrophages were treated with *METTL3* or control siRNA ($n = 4$ cells). Renilla luciferase activity was measured and normalized to firefly luciferase activity. **m** *Ptx3* mRNA stability in WT and *Mettl3* KO BMDMs treated with actinomycin D at the indicated times point determined by RT-qPCR ($n = 3$ animals). Statistical analysis of the data was performed using two-sided unpaired *t* test with Welch's correction (**j** left) or not (**d**, **h**, **j** right, **k**), or 2-way ANOVA (**i**, **l**, **m**) followed by Sidak's multiple comparison tests. Data are presented as means ± SEM from one of three independent experiments. *$P < 0.05$, **$P < 0.01$, ***$P < 0.001$, ****$P < 0.0001$; n.s, not significant.

imply that PTX3 may be a potential biomarker for the diagnosis and assessment of childhood allergic asthma.

## Discussion

m⁶A mRNA modification emerges as an important regulator of diverse biological processes involved in various disease states. In this study, we demonstrated that m⁶A modification could regulate the progression of allergic asthma by influencing macrophage homeostasis. In brief, *METTL3* was lowly expressed in monocyte-derived macrophages from childhood allergic asthma, while an inverse correlation between *METTL3* expression and disease severity was identified. Conditional knockout of *Mettl3* in myeloid cells enhanced Th2 cell response and aggravated allergic airway inflammation by facilitating M2 macrophage activation. Gain and loss of function studies revealed that METTL3 acted as a negative regulator of M2 macrophages, partly through the suppression of PI3K/AKT and JAK/STAT6 signaling. Mechanistically, using m⁶A-seq and RNA-seq, we verified that METTL3 inhibited M2 macrophage activation by promoting the degradation of *PTX3* mRNA in a YTHDF3-mediated, m⁶A-dependent manner. Silencing *Ptx*3 significantly alleviated allergic inflammation in *Mettl3* KO mice, whereas higher PTX3 expression in plasma was positively associated with airway inflammation in childhood allergic asthma. Notably, the regulatory effect of the METTL3/YTHDF3-m⁶A/PTX3 axis on macrophage homeostasis was also extended to its role in autophagy maturation via regulating *STX17* expression. Collectively, this study highlights the critical role of METTL3 deficiency in the pathogenesis of allergic asthma airway inflammation, as featured by promoting M2 macrophage activation and enhancing Th2 response, and uncovers a signaling axis involving METTL3/YTHDF3-m⁶A/PTX3/STX17 in macrophage activation and autophagy maturation (Fig. 8f).

Altered m⁶A modification has been reported in various diseases, including inflammatory conditions. Our work demonstrated an association between decreased *METTL3* levels in monocyte-derived macrophages and the severity of childhood allergic asthma. Combined with neutrophils and macrophage depletion in vivo, we confirmed that *Mettl3* deficiency in myeloid cells aggravated airway inflammation via macrophages. An increasing number of studies demonstrated a positive correlation between asthma severity and elevated M2 macrophage numbers[10–13]. Notably, recent evidence suggests that METTL3 could facilitate or repress the activation of M1 macrophages depending on the local immune microenvironment[25,26,45], indicating that m⁶A modification may play important role in macrophage homeostasis. In this study, through a combination of RNA-seq, flow cytometry, and immunofluorescence, we observed that M2 activation-associated markers were significantly enhanced in *Mettl3*-deficient macrophages and lung tissues, accompanied by the activation of PI3K/AKT and JAK/STAT6 signaling. Allergic asthma is often considered a Th2 cell-mediated chronic immune response, while M2 macrophages could

release chemokines such as CCL-17 and CCL-22 to recruit the Th2 cells and amplify Th2 cell responses in the bronchial tissue[32]. Here, we revealed that *Mettl3* deficiency in mice highly promoted Th2 cell responses in lung tissues via facilitating M2 macrophage activation, implying the importance of METTL3 in allergic airway inflammation.

MeRIP-seq data revealed that the 3′UTR regions of *PTX3* transcripts contained m⁶A modifications, and eliminating these retarded the degradation of *PTX3* mRNA. PTX3, a member of the pentraxin family, is primarily produced by dendritic cells, macrophages, and endothelial cells in response to inflammation. It can have potentially conflicting roles in the regulation of inflammation, antimicrobial resistance, and disease pathogenesis, depending on tissue context, cellular source, and abundance[34,44,46]. PTX3 affects inflammation through multiple mechanisms, such as interaction with complement components, Fcγ receptors, and pathogenic moieties in microbes[47]. The current study demonstrated that *PTX3* depletion significantly suppressed M2 macrophage activation through PI3K/AKT and JAK/STAT6 signaling, consistent with previous reports documenting that PTX3 reduces M1 and increases M2 macrophage activation[34,35]. Furthermore, it has been suggested that plasma PTX3 levels could act as a biomarker in various lung diseases, such as lung carcinoma[43], chronic obstructive pulmonary disease (COPD)[48], and acute lung injury (ALI)[49], while sputum PTX3 levels represent a candidate biomarker for the evaluation of airway inflammation and remodeling in children with asthma[50]. Interestingly, a recent study identified that exogenous PTX3 could exacerbate asthma, by promoting eosinophil infiltration into the lung[51]. However, another independent study found that *Ptx3* deficiency in an OVA-induced mouse asthma model resulted in augmented AHR, mucus production, and IL-17-dominant airway inflammation[52]. In this study, we detected elevated levels of serum PTX3 in CRE-treated *Mettl3* KO mice. We observed that *Ptx3* deficiency markedly attenuated *Mettl3* knockout-induced airway inflammation in allergic asthma by suppressing M2 macrophage activation, but the effect of PTX3 on the other cells in vivo can not be excluded. Noticeably, our data also uncovered that plasma levels of PTX3 were higher in childhood allergic asthma patients, showing a positive correlation with disease severity. These findings identify PTX3 as a key regulator of M2 macrophage activation and a potential biomarker in allergic asthma.

m⁶A modification accomplishes its biological function primarily by recruiting various RNA-binding proteins, which subsequently direct multi-protein complexes to selectively bind to m⁶A-containing transcripts. We detected a robust decrease of *YTHDF3* levels in *METTL3*-deficient macrophages, and in monocyte-derived macrophages from children with allergic asthma. It appears that METTL3 regulates M2 macrophage activation in a YTHDF3-dependent manner. Cytoplasmic 'readers', such as YTHDF1 and YTHDF2, were demonstrated to modulate the degradation and translation of m⁶A-modified mRNAs[16,18]. Although the cytoplasmic m⁶A 'reader' YTHDF3 has been reported to

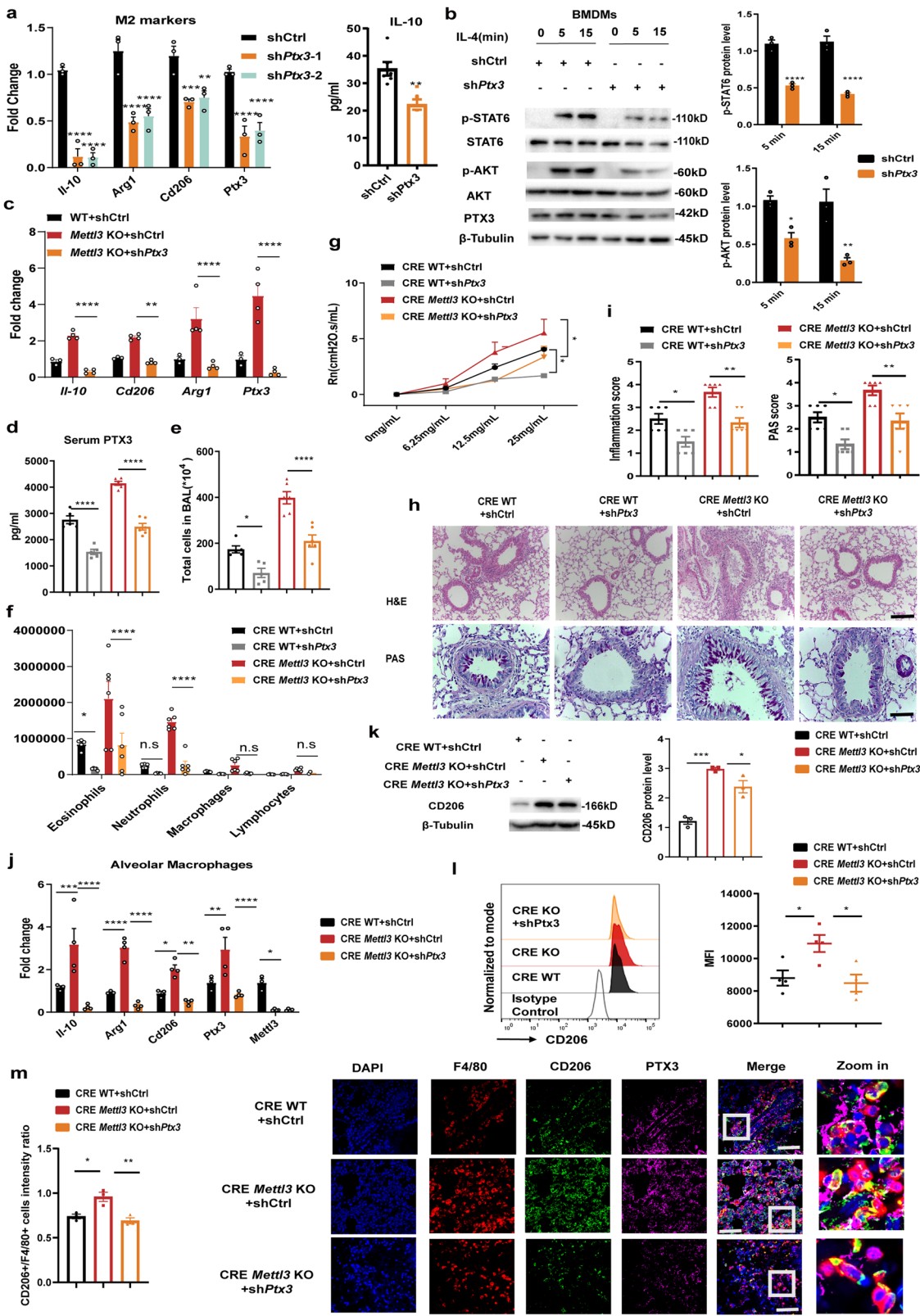

promote mRNAs translation[16,19], increasing evidence indicates that YTHDF3, through cooperation with YTHDF1 and YTHDF2, affects methylated mRNAs decay, by recruiting different complexes to regulate mRNAs clearance[37,38,53,54]. Here, we provide the evidence that the METTL3/YTHDF3 m[6]A axis preferentially affects the 3'UTR of *PTX3* when triggering its decay during M2 macrophage activation. Mutations eliminating the RRACH motif can partially protect against this

m[6]A-regulated degradation. The above findings suggest that the control of *PTX3* mRNA decay by the METTL3/YTHDF3 m[6]A axis represents the mechanism for suppressing M2 macrophage activation.

Autophagy is an evolutionarily conserved process of recycling cytosol components and maintaining intracellular homeostasis. Accumulating evidence indicates that autophagy is essential for M2 macrophage activation. For example, impaired autophagy inside

**Fig. 5 | PTX3 is a functionally essential target of METTL3-modulated macrophage activation in allergic asthma. a** Knockdown of *Ptx3* in BMDMs using two distinct shRNAs. The M2-associated makers expression in IL-4–stimulated BMDMs was quantified by RT-qPCR (left) (*n* = 3 cells). The levels of IL-10 secretion were detected by ELISA (right) (*n* = 6 cells for control group and *n* = 4 cells for knockdown group). **b** Western blot analysis of STAT6 and AKT phosphorylation in BMDMs transfected with *Ptx3* or Ctrl shRNA, after IL-4 stimulation (*n* = 3 cells). **c** RT-qPCR showing M2 activation-associated markers expression in BMDMs from WT and *Mettl3* KO mice, with or without *Ptx3* knockdown (*n* = 3 cells for WT group and *n* = 4 cells for other groups). **d** Depletion of *Ptx3* in CRE–induced allergic asthma model. Serum PTX3 levels were determined by ELISA. **e** Total and (**f**) differential BALF cell numbers from experimental animals were analyzed by flow cytometry (*n* = 5 animals for WT groups and *n* = 6 animals for KO groups). **g** Assessment of methacholine-induced AHR in mice (*n* = 4 animals for WT groups and *n* = 5 animals

for KO groups). **h** Histopathological changes in the lung tissues were examined by H&E- and PAS-staining (scale bars: 200 μm and 100 μm, respectively). **i** Calculated inflammation and PAS scores (*n* = 6 animals). **j** RT-qPCR (*n* = 3 cells for WT groups and *n* = 4 cells for KO groups) and (**k**) Western blot (*n* = 3 cells) assays detecting M2-associated markers expression in alveolar macrophages purified from experimental animals. **l** Flow cytometry analysis of the MFI of the M2 macrophage subpopulation in BALF (*n* = 4 animals). **m** Representative composite fluorescent images showing the spatial localization of M2 macrophages (F4/80⁺, red; CD206⁺, green) and PTX3 levels (purple) in lung tissues. Scale bars: 25 μm. Quantification analysis was performed (*n* = 3 animals). Statistical analysis of the data was performed using two-sided unpaired *t* test (**a** right), 1-way ANOVA (**d, e, i, k, l, m**), and 2-way ANOVA (**a** left, **b, c, f, g, j**) followed by either Tukey's or Sidak's multiple comparison tests. Data are presented as means ± SEM from one of three independent experiments. *$P < 0.05$, **$P < 0.01$, ***$P < 0.001$, ****$P < 0.0001$; n.s, not significant.

macrophages increased inflammation in obese mice by promoting M1 and decreasing M2 activation[41]; CCL2 and IL-6 are potent inducers of autophagy in these cells, and they can trigger development along with the M2 phenotype[42]; A knockdown of lincRNA-Cox2 in macrophages inhibited NLRP3 inflammasome activation and promoted autophagy, decreasing pro-inflammatory cytokine secretion[55]; In the present study, KEGG pathway analysis revealed that molecules associated with autophagy were enriched in both *PTX3* and *METTL3*-deficient macrophages. Moreover, METTL3/YTHDF3-m⁶A/PTX3 axis exerted its regulatory effects on autophagy maturation, which is a critical but often overlooked step of autophagy. A recent report showed that a set of SNARE complex proteins, including STX17, SNAP29, and VAMP8, are essential for autophagy maturation. The depletion of STX17 causes autophagosome accumulation and prevents their further degradation[40]; The macrophage-specific V-ATPase subunit, ATP6V0D2, facilitates autophagy maturation and restricts macrophage inflammation through its interaction with STX17[39]. Here, we verified that the METTL3/YTHDF3-m⁶A/PTX3 axis played a vital role in the autophagy maturation of macrophage by regulating STX17 expression. Notably, through RNA-seq analysis, we also found that *PTX3*-deficient macrophages reduced the levels of key transcription factors such as *ATF3* and *MYC*, and histone acetyltransferase *KAT6A*, which have been reported to modulate autophagy-associated gene expression[56,57]. Our observations demonstrate a strong link between autophagy and macrophage homeostasis mediated by the METTL3/YTHDF3-m⁶A/PTX3 axis and its downstream targets. A detailed, comprehensive investigation into how the PTX3/STX17 axis modulates autophagy and macrophage homeostasis in allergic asthma is certainly needed.

A limitation of this study is that the detailed function of PTX3 has not been verified in *Ptx3* conditional KO mice. In addition, although our studies with monocyte-derived macrophages in children with allergic asthma and myeloid cells in animal models have suggested a crucial role for METTL3/m⁶A modification in the development of allergic asthma mediated by macrophage activation, lung tissue-resident macrophages evidence for this phenotype in childhood asthma is lacking, warranting further studies. Importantly, large cohorts of childhood allergic asthma need to be constructed, which evaluate the possibility of *METTL3* or *PTX3* levels as potential biomarkers for the diagnosis and assessment of childhood allergic asthma.

In summary, we provide compelling in vitro and in vivo evidence demonstrating the critical role of m⁶A modification in controlling macrophage homeostasis in allergic asthma. The combined network of 'writer' METTL3, 'reader' YTHDF3, and 'target' PTX3 molecules highlight an intriguing m⁶A-dependent gene regulatory mechanism. The insights gained in this study will help to further our understanding of the pathophysiological roles of m⁶A modification in macrophage homeostasis, and help develop therapeutic strategies for the treatment of allergic asthma.

## Methods

### Human study design
55 children with allergic asthma and 50 broadly age- and sex-matched healthy controls were recruited from the Children's Hospital of Fudan University. Informed consent was obtained from the parents of all participants. The patients with asthma were diagnosed according to the 2016 edition of the Guidelines for the Diagnosis and Prevention of Childhood Bronchial Asthma of China[5]. Individuals without a history of asthma and any other allergic diseases acted as controls. Detailed characteristics of study participants are presented in Supplementary Table 2. The study was approved by the Research Ethics Board of the Children's Hospital of Fudan University (No. 2020-81).

Human PBMCs were isolated using Ficoll-Paque density gradient solution (density=1.077 g/ml, GE Healthcare). Plasma was isolated from peripheral blood by centrifugation at 400 g for 10 min. Peripheral blood was mixed with PBS and overlaid on top of Ficoll. Following centrifugation, PBMCs were aspirated from the Ficoll-plasma interface and washed twice in PBS.

Human monocyte-derived macrophages were isolated as previously described with some minor modifications[58]. Briefly, CD14⁺ monocytes were purified from the PBMCs by magnetic cell sorting using CD14 microbeads (Miltenyi Biotech). To trigger macrophage differentiation, CD14⁺ monocytes were cultured in a complete medium, consisting of RPMI-1640 medium, 10% heat-inactivated fetal bovine serum (FBS, Gibco), and 20 ng/ml granulocyte-macrophage colony-stimulating factor (GM-CSF, R&D Systems). Macrophages were harvested after 7 days.

### Mice
*Mettl3*^flox/flox^ mice were generated by Shanghai Model Organisms Center, Inc (China), using a LoxP targeting system. Briefly, the *Mettl3* targeting vector was designed to flank exon 2–3 with LoxP sites and electroporated into C57BL/6 embryonic stem cells (ES). Positive ES clones were confirmed by PCR and sequencing, and injected into C57BL/6 blastocysts to generate chimeric offspring. Chimeric mice were mated with C57BL/6 mice to obtain *Mettl3*^flox/flox^ mice. These *Mettl3*^flox/flox^ mice were crossed with *Lyz2*-Cre mice (The Jackson Laboratory) to generate *Mettl3*^fl/fl^ (WT) and *Mettl3*^fl/fl^*Lyz2*^Cre/+^ (KO) mice. Primers used to identify genetically modified mice are listed in Supplementary Table 3. All mice were housed, bred, and maintained under specific pathogen-free conditions, fed standard laboratory chow, and kept on a 12 h light/dark cycle and temperature and humidity were kept at 22 ± 1 °C, 55% ± 5%. Experiments complied with the relevant laws and institutional guidelines, as overseen by the Animal Studies Committee of the Children's Hospital of Fudan University.

### Cockroach allergen-induced mouse asthma model
The mouse asthma models were conducted in female mice since female mice are more susceptible to the development of allergic airway inflammation than male mice[59–61]. A cockroach allergen-induced

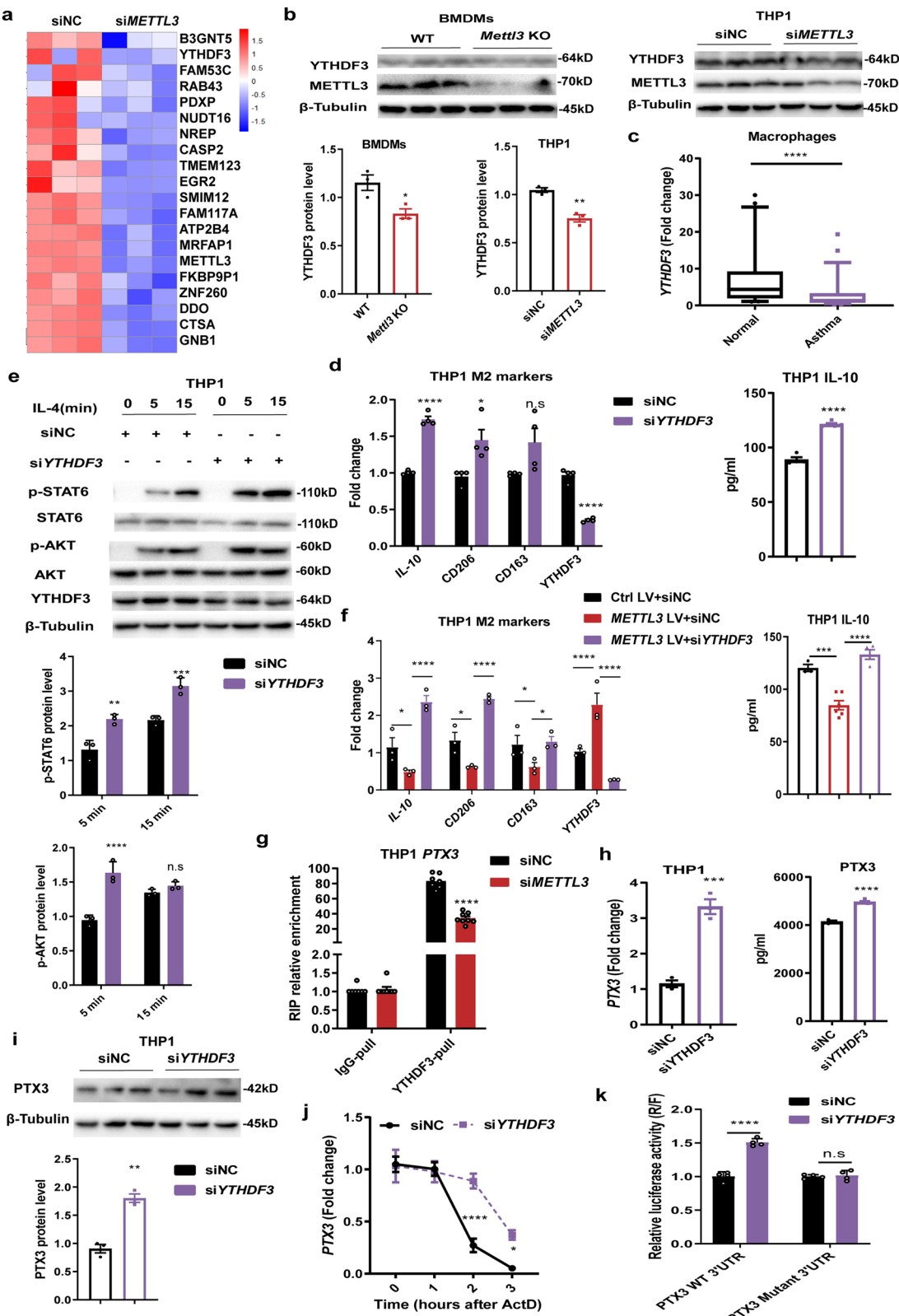

asthma mouse model was established as described in our previous work[12,13]. Briefly, six- to eight-week-old female WT and *Mettl3* KO mice were sensitized by intratracheal inhalation of 20 μg cockroach extract (CRE, GREER Laboratories) per mouse on days 1, 2, 3, 14, and challenged on days 20 and 22 with the same amount of CRE. Control mice received PBS during the sensitization and challenge phases. On day 23, mice were euthanized by $CO_2$ asphyxiation, and lung tissues were dissected and analyzed. Blood was collected to screen for serum PTX3

levels. To produce *Ptx3* knock-down animals, $5 \times 10^6$ plaque-forming units of sh*Ptx3* lentivirus or control (shCtrl) virus were administered by intra-tracheal instillation on day 14.

### Flow cytometry
Cells were stained with Fixable Viability Stain 780 (BD Biosciences, 565388, 1:1000) and then incubated with an anti-CD16/32 monoclonal antibody (eBioscience, 14-0161-82, 1:100). Analysis of bronchoalveolar

**Fig. 6 | METTL3/YTHDF3 m⁶A axis degrades *PTX3* mRNA during M2 macrophage activation. a** Heatmap profiling downregulated genes in human *METTL3*-deficient THP1-derived macrophages. **b** Western blot showing protein levels of YTHDF3 in both *Mettl3* KO mice BMDMs (*n* = 3 animals) and *METTL3*-deficient THP1-derived macrophages (*n* = 3 cells), respectively. **c** RT-qPCR analysis of the abundance of *YTHDF3* in monocyte-derived macrophages from 55 childhood allergic asthma and 50 healthy controls. The detailed minimum, median, maximum, 25th, 75th percentile (box), and 5th and 95th percentile (whiskers) of box plots were provided in the Source data file. **d** RT-qPCR (left) analysis of M2-associated markers mRNA levels (*n* = 4 cells), and ELISA (right) detection of the IL-10 secretion levels (*n* = 4 cells for control group and *n* = 5 cells for knockdown groups) in THP1-derived macrophages transfected with *YTHDF3* siRNAs pools (200 nM), followed by IL-4 stimulation. **e** Elevated levels of p-AKT and p-STAT6 in *YTHDF3*-deficient THP1-derived macrophages were detected by Western blot (*n* = 3 cells). **f** RT-qPCR (*n* = 3 cells) and ELISA (*n* = 3, 6, 4 cells for Ctrl LV+siNC, *METTL3* LV+siNC, *METTL3* LV +si*YTHDF3* group, respectively) showing M2 activation-associated markers

expression in THP1-derived macrophages overexpressing *METTL3*, with or without *YTHDF3* knockdown. **g** YTHDF3 RIP-analysis of *PTX3* m⁶A level in control or *METTL3*-deficient THP1-derived macrophages (*n* = 7 cells for control group and *n* = 8 cells for knockdown group). Relative enrichment was normalized by input. **h** The increase of *PTX3* mRNA (*n* = 3 cells) and secretion levels (*n* = 3 cells for control group and *n* = 5 cells for knockdown group) in *YTHDF3*-deficient THP1-derived macrophages was quantitated by qPCR and ELISA. **i** Western blot showing up-regulated PTX3 protein levels in *YTHDF3*-deficient THP1-derived macrophages (*n* = 3 cells). **j** Effect of *YTHDF3* knockdown on *PTX3* mRNA stability in THP1-derived macrophages treated with actinomycin D (*n* = 4 cells). **k** Dual-luciferase reporter assays demonstrating the effect of YTHDF3 on *PTX3* 3′UTR reporters with either WT or mutated m⁶A site (*n* = 4 cells). Statistical analysis of the data was performed using two-sided unpaired *t* test (**b, d, h, i**), Mann–Whitney test (**c**), 1-way ANOVA (**f** right), and 2-way ANOVA (**e, f** left, **g, j, k**) followed by either Tukey's or Sidak's multiple comparison tests. Data are presented as means ± SEM from one of three independent experiments. **P* < 0.05, ***P* < 0.01, ****P* < 0.001, *****P* < 0.0001; n.s, not significant.

lavage fluid (BALF) was performed as described previously[12]. Briefly, cells were stained for eosinophils, macrophages, neutrophils, and lymphocytes. Eosinophils were defined as SSC^high SiglecF⁺ Mac-3⁻ cells, alveolar macrophages were identified as SSC^high SiglecF⁺ Mac-3⁺ cells, neutrophils were recognized as SSC^high Gr-1⁺cells, and lymphocytes were identified as FSC^low/ SSC^low cells expressing CD3 or CD19 (BioLegend: SiglecF-PE, 155505, 1:100; MAC-3-FITC, 108504, 1:200; Gr-1-APC, 108412, 1:200; CD19-PerCP/Cy5.5, 152405, 1:250; CD3-PerCP/Cy5.5, 100327, 1:100). For M2 macrophage subpopulation analysis, BALF cells were further purified by macrophage adherence and stimulated by IL-4 for 12 h. The cells were collected and stained with anti-CD11c, F4/80, and CD206 antibodies (BioLegend: CD11c-FITC, 117305, 1:100; F4//80-PE, 123109, 1:200; CD206-APC, 141707, 1:40). For Th1/Th2-associated intracellular cytokine staining, cells isolated from mediastinal lymph nodes (MLNs) were stimulated with phorbol myristate acetate (PMA) and ionomycin for 8 h in the presence of brefeldin A and monensin (ebioscience). Cells were fixed, permeabilized, washed, and stained with anti-CD3, CD4, IFN-gamma and IL-4 antibodies (BioLegend: CD3-FITC, 100203, 1:50; CD4-PerCP/Cy5.5, 100433, 1:80; IL-4-APC, 504105, 1:20; IFN-gamma-PE, 163503, 1:40). For the gating of intracellular cytokines markers, cells were incubated with isotype control and various cytokine antibodies, respectively. The isotype control plot was used to set the gates of intracellular cytokines. Data were analyzed with FlowJo V10 software. The gating strategy is provided in Supplementary Fig. 13.

### Pulmonary function testing for assessing airway hyperresponsiveness (AHR)

We have previously described the assessment of AHR in mice with induced asthma in detail[13]. In brief, mice were anesthetized with phenobarbital sodium (50 mg/kg body weight), and PBS or increasing concentrations of aerosolized methacholine were nebulized into airways. Invasive measurements of AHR were made on a FinePointe Resistance and Compliance System (Buxco Research Systems). Airway resistance was calculated using FinePointe software. The normalization function of the software was used, and data were represented as a deviation from 0 mg/ml.

### Isolation of primary neutrophils and depletion of neutrophils

Mouse primary neutrophils were isolated from the bone marrow of mice using the EasySep Mouse Neutrophil Enrichment Kit (STEMCELL) according to the manufacturer's instructions. For neutrophil depletion, every 72 h during CRE treatment, mice were i.p. injected with 200 μg anti-Ly6G Ab (BioLegend, 1A8) or control IgG2a Ab diluted in sterile PBS[28]. For confirmation of depletion during allergic asthma, BALF was harvested and analyzed by flow cytometry for a reduction in CD45⁺CD11b⁺Gr1⁺ cells (BioLegend: CD45-APC, 103111, 1:100; CD11b-FITC, 101205, 1:200; Gr-1-PE, 127607, 1:200).

### Depletion of macrophages

Macrophages were depleted using CLs (ClodronateLiposomes.org) as described previously[28]. Briefly, mice were anesthetized and i.p. injected with 200 μl clodronate liposomes every 72 h during CRE treatment. Controls included i.p injections of 200 μl PBS. For confirmation of depletion during allergic asthma, BALF was harvested and analyzed by flow cytometry for a reduction in SiglecF⁺Mac-3⁺ cells.

### Immunohistochemistry and immunofluorescence

Sections (5 μm) were prepared from formalin-fixed paraffin-embedded (FFPE) lungs, and stained for hematoxylin and eosin (H&E) and periodic acid–Schiff (PAS). An inflammation score on a scale of 0 to 4 was assigned to the perivascular and peribronchial areas of multiple fields per slide, based on previously published methodology[62]. A score of 0 represented a complete lack of inflammation, and a designation of 4 was assigned to areas where inflammatory cells were present at a depth of greater than 3 cells completely surrounding an airway or blood vessel. The numerical scores for the abundance of PAS-positive mucus-containing cells in each airway were determined as follows: 0, <0.5% PAS-positive cells; 1, 5–25%; 2, 25–50%; 3, 50–75%; 4, >75%. For the detection of neutrophils, FFPE tissue slides were stained with anti-Gr1 (Cell Signaling Technology, #87048, 1:50).

Lung immunofluorescent staining was performed using a Multiple fluorescent immunohistochemistry staining Kit (Absin, Shanghai, China), according to the manufacturer's instructions. 5 μm FFPE tissue slides were deparaffinized, rehydrated, and fixed. Antigen retrieval was performed using microwave treatment (MWT), and the sections were incubated with rabbit anti-F4/80 (Cell Signaling Technology, #70076, 1:250), anti-CD206 (Abcam, ab64693, 1:200), and anti-PTX3 (Proteintech, 13797-1-AP, 1:100) antibodies. Incubation with HRP-labeled anti-rabbit polymer was performed, followed by TSA opal fluorophores (TSA 520, TSA 570, or TSA 650) incubation. MWT was performed at each cycle of staining to remove the Ab/ TSA complex. Finally, all slides were counterstained with DAPI and scanned using a Leica TSC SP8 confocal microscope.

### Isolation of bone marrow-derived macrophages (BMDMs)

Mouse BMDMs were isolated and cultured as described previously[12]. Briefly, bone marrow was extracted from femurs and, after erythrocyte lysis, plated in complete medium, consisting of DMEM containing 10% FBS and 20 ng/ml macrophage colony-stimulating factor-1 (M-CSF, R&D System). Cells were allowed to differentiate for 7–8 days before use.

### Isolation of alveolar macrophages

Isolation of mouse alveolar macrophages was performed according to our previously published work with some modifications[12,63]. Lung tissues harvested from WT and *Mettl3* KO mice were minced and

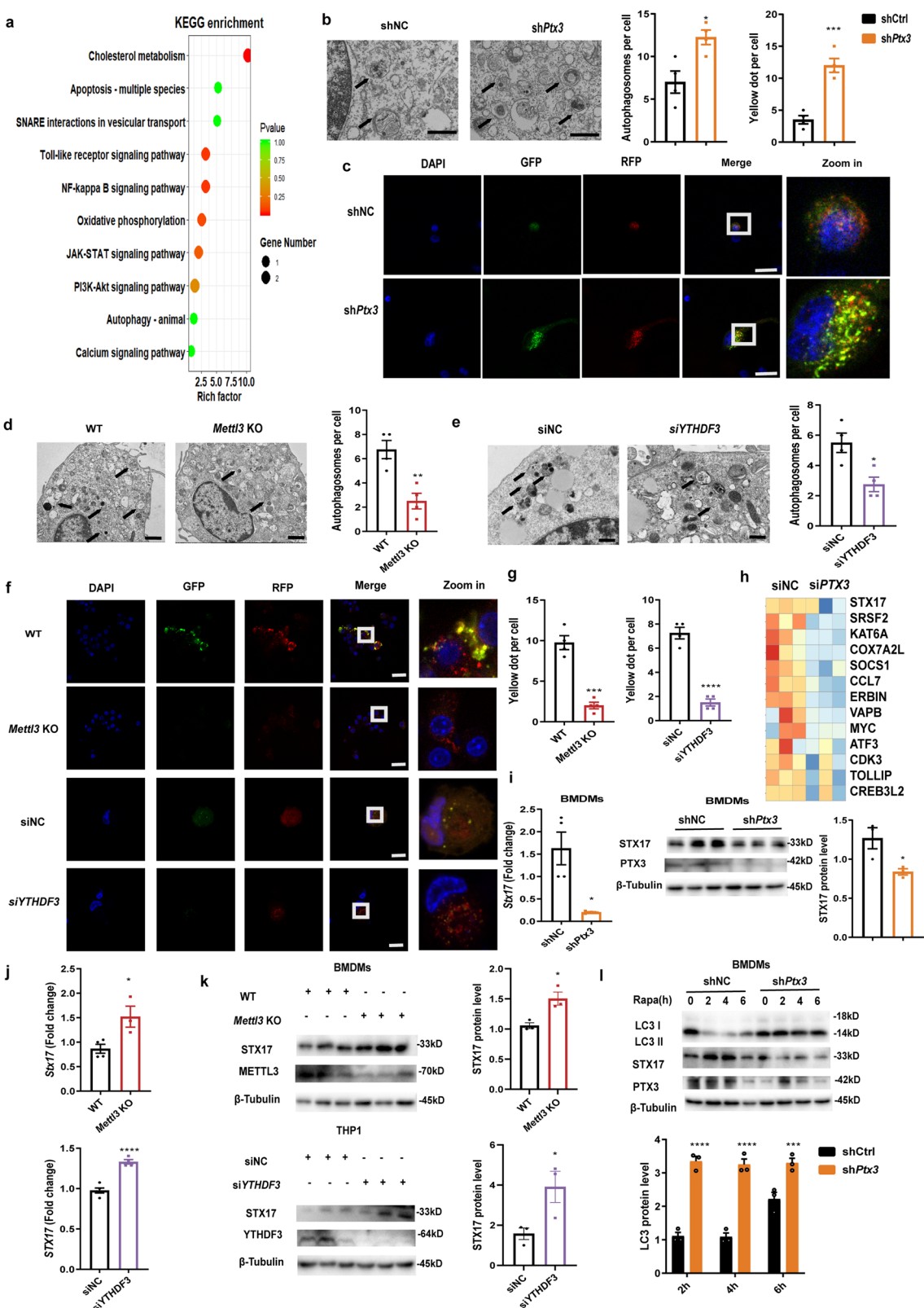

collected into gentleMACS C Tubes, containing the enzyme mix of the Lung Dissociation Kit (Miltenyi Biotech). Tissues in C Tubes were subjected to the run program 37 °C_m_lung for 30 min on the gentleMACS Octo Dissociator. Cell suspensions were passed through a 70 μm MACS SmartStrainer (Miltenyi Biotech), followed by centrifugation at 300 g for 7 min at 4 °C. All isolated cells were further purified by EasySep Mouse F4/80 Positive Selection Kit (STEMCELL) and

macrophage adherence, and stimulated by IL-4 for 12 h. The percentage of alveolar macrophages in purified cell populations was detected by flow cytometry analysis using anti-CD11c and F4/80 antibodies.

### Cell culture, differentiation, and transfection
Human THP1 cells (obtained from ATCC) were maintained in RPMI-1640 medium containing 10% heat-inactivated FBS. To generate

**Fig. 7 | METTL3/YTHDF3-m⁶A/PTX3/STX17 axis controls autophagy maturation in macrophages. a** Human THP1-derived macrophages were infected with *PTX3* siRNA pools (200 nM) or control, followed by IL-4 stimulation. KEGG analysis of pathways enriched in *PTX3*-deficient macrophages. **b** Transmission electron microscopy (TEM) determined the number of autophagosomes in control and *PTX3*-knockdown BMDMs ($n = 4$ cells). Representative images were shown together with mean values of the number of autophagosomes per cell. Scale bars, 1 μm. **c** Control and *PTX3*-deficient BMDMs were transiently transfected with mRFP-GFP-LC3-expressing adenovirus, followed by autophagy induction. The LC3 puncta were analyzed by confocal microscopy, and the mean number of LC3 puncta (top) was determined by counting ($n = 4$ cells). Scale bars: 10 μm. TEM demonstrates the decreased autophagosomes in *Mettl3* KO BMDMs (**d**) ($n = 4$ animals) and *YTHDF3*-deficient THP1-derived macrophages (**e**) ($n = 4$ cells). Scale bars, 2 μm, and 1 μm, respectively. **f, g** The autophagy flux analysis showing the number of LC3 puncta in *Mettl3* KO BMDMs ($n = 4$ animals) and *YTHDF3*-deficient THP1-derived macrophages ($n = 4$ cells). Scale bars, 25 μm, and 10 μm, respectively. **h** Heatmap identifying the down-regulated transcripts in *PTX3*-deficient THP1-derived macrophages. **i** RT-qPCR ($n = 4$ cells) and immunoblot ($n = 3$ cells) verifying reduced levels of STX17 in *PTX3*-knockdown BMDMs. **j** RT-qPCR showing up-regulated *STX17* expression in *Mettl3* KO BMDMs ($n = 4$ animals for WT group and $n = 3$ animals for KO group) and *YTHDF3*-deficient THP1-derived macrophages ($n = 5$ cells for control group and $n = 4$ cells for knockdown group), respectively. **k** The increased STX17 levels in *Mettl3* KO BMDMs ($n = 3$ animals) and *YTHDF3*-deficient THP1-derived macrophages ($n = 3$ cells) by Western blot. **l** Immunoblot analysis of LC3, STX17, and PTX3 levels in control and *PTX3*-knockdown BMDMs treated with rapamycin for 0–6 h ($n = 3$ cells). Statistical analysis of the data was performed using two-sided unpaired *t* test with Welch's correction (**i** left) or not (**a, b, d, e, g, i** right, **j, k**), and 2-way ANOVA (**l**) followed by Sidak's multiple comparison tests. Data are presented as means ± SEM from one of three independent experiments. *$P < 0.05$, **$P < 0.01$, ***$P < 0.001$, ****$P < 0.0001$.

macrophage-like cells, the seeded THP1 cells were treated with 200 nM PMA (Sigma-Aldrich) for 48 h. For M1 and M2 activation, macrophages were stimulated with LPS (200 ng/ml, 3 h, Sigma-Aldrich) or IL-4 (20 ng/ml, 24 h, R&D Systems), respectively. Small interfering RNAs (siRNAs), including *PTX3* shRNA lentivirus (*PTX3* shRNA), and *METTL3*-overexpressing lentivirus (*METTL3* LV) were synthesized by Gene-Pharma (Shanghai, China). The sequences of siRNAs and controls are provided in Supplementary Table 4. THP1 cells were transfected with these RNAs using Lipofectamine RNAiMAX (Invitrogen), according to the manufacturer's instructions.

## m⁶A-seq assay and data analysis

*METTL3*-knockdown human THP1-derived macrophages were incubated with IL-4 for 24 h. Total RNA was isolated and mRNAs with polyA were enriched using Oligo-dT. RNA fragmentation reagent (10 mM ZnCl₂, 10 mM Tris−HCl pH 7.0) was used to randomly fragment RNA, and an m⁶A antibody (Synaptic Systems) was applied for m⁶A pull-down (m⁶A-IP). Both input and m⁶A IP samples were prepared for next-generation sequencing. The library preparation was constructed using the TruSeq Stranded mRNAs Sample Prep Kit (Illumina), and then was deeply sequenced on the Illumina Hiseq 4000 at OE Biotech. Co. Ltd (Shanghai, China).

For data analysis, the reads from input and m⁶A IP samples were aligned to the human genome GRCh38 using HISAT2 with default parameters. The differentially expressed transcripts of m⁶A methylome between *METTL3*-knockdown and Ctrl macrophages were detected by MeTDiffpeak calling software. Sequence motifs enriched in m⁶A peak regions compared to control regions were identified using MEME. Integrative Genomics Viewer (IGV) was used to visualize the distribution of the m⁶A peaks.

## RNA-seq assay and data analysis

Total RNA was isolated from *METTL3* or *PTX3*-knockdown THP1-derived macrophages stimulated by IL-4 for 24 h. The TruSeq RNA Sample Preparation Kit (Illumina) was used for library preparation. Libraries at a final concentration of 10 pM were clustered onto a single read flow cell by cBot and sequenced to 150 bp using an Illumina NovaSeq 6000 instrument (Illumina). Library construction and sequencing were performed by Sinotech Genomics. Co. Ltd (Shanghai, China). Differential genes expression were analyzed by the standard Illumina sequence analysis pipeline. To comprehensively analyze the RNA-seq data, filtering criteria were applied (≥1.5-fold change, $P < 0.05$).

## Quantitative real-time PCR

Total RNA was extracted from cells using Trizol Reagent. cDNA was synthesized using the PrimeScript RT Reagent Kit (Takara). To quantify mRNA expression, cDNAs were amplified by RT-qPCR using the SYBR Premix Ex Taq RT-PCR kit (Takara). *ACTIN* levels were used as internal housekeeping controls in both human and mouse samples. The data was collected using Roche LC 480. Fold-change in expression was calculated using the $2^{-\Delta\Delta Ct}$ method. The primer sequences for all genes are provided in Supplementary Table 3.

## Western blot analysis

Lysates were resolved by electrophoresis, transferred to PVDF membranes, and probed with antibodies directed against phosphorylated (Ser473) AKT (#4060, 1:1000), AKT (#2920, 1:1000), phosphorylated (Tyr641) STAT6 (#9361, 1:1000), STAT6 (#9362, 1:1000), METTL3 (#96391, 1:1000) (all from Cell Signaling Technology), PTX3 (13797-1-AP, 1:1000), YTHDF3 (25537-1-AP, 1:1000), LC3 (14600-1-AP, 1:1000), STX17 (17815-1-AP, 1:1000) (all from Proteintech), CD206 (ab64693, 1:1000) and β-Tubulin (ab6046, 1:5000) (all from Abcam). All results were normalized to those of β-Tubulin, which was used as a loading control. The data was collected using Bio-Rad ChemiDoc XRS⁺. Representative figures from three biological experiments were shown.

## Enzyme-linked immunosorbent assay (ELISA)

The protein levels of IL-10, TNF, IL-4, IL-13, and PTX3 were measured in culture medium, serum, and plasma with the DuoSet ELISA kit (R&D Systems) according to the manufacturer's instructions.

## The SELECT detection assay

The SELECT was performed using the Epi-SELECT m⁶A site identification Kit (Epbiotek, Guangzhou, China) as described previously[64]. In brief, 1 μg total RNA was mixed with 1 μM Up Primer, 1 μM Down Primer, and 2 μl dNTP in 2 μl 10× Reaction buffer. Annealing of RNA and primers was performed for 1 min at each 90 °C, 80 °C, 70 °C, 60 °C, 50 °C, and 6 min at 40 °C. Subsequently, the former mixture, DNA polymerase, ligase, and ATP were added in a total volume of 20 μl and incubated for 20 min at 40 °C and 80 °C. The qPCR reaction was performed using SYBR Premix. Relative SELECT products were calculated by normalization to the RNA abundance determined by the control. The primer sequences are provided in Supplementary Table 3.

## MeRIP-qPCR

The Magna MeRIP m⁶A Kit (Millipore) was used to perform MeRIP-qPCR according to the manufacturer's instructions. 200 μg of total RNA was extracted from THP1-derived macrophages infected with *METTL3* siRNA or control siRNA. RNA was sheared to approximately 100 nt in length and incubated with m⁶A antibody (Millipore, MABE1006, 2 μg/ml)-or IgG-conjugated Protein A/G Magnetic beads at 4 °C overnight. Methylated RNAs were immunoprecipitated with beads, eluted, and recovered with the RNeasy kit (QIAGEN). One-tenth of fragmented RNA was saved as input control and further analyzed by RT-qPCR along with the MeRIPed RNAs using primers listed in Supplementary Table 3. The related enrichment of m⁶A in each sample was

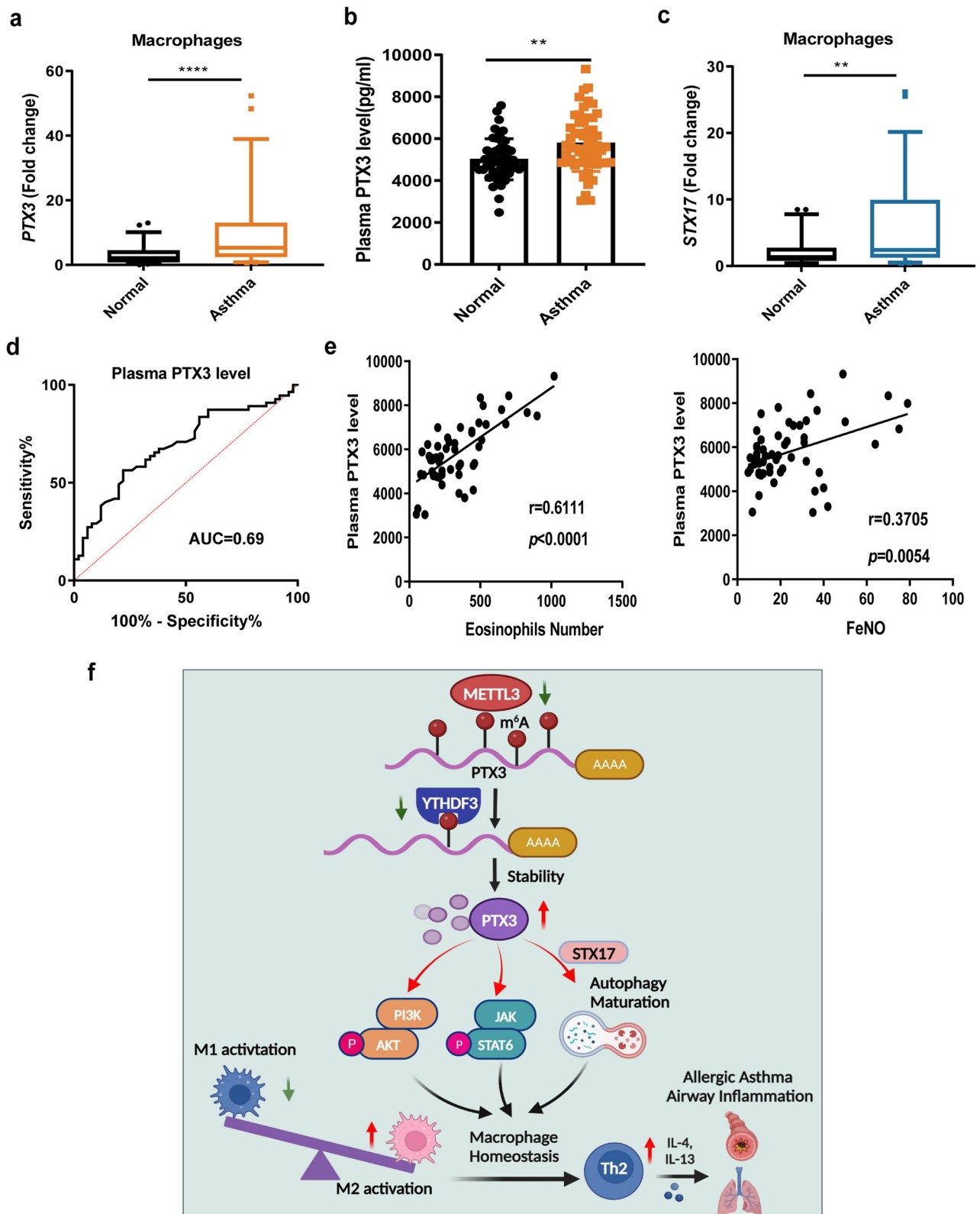

**Fig. 8 | Enhanced levels of PTX3 are associated with disease severity of childhood allergic asthma.** The *PTX3* mRNA levels in monocyte-derived macrophages (**a**) and protein levels in the plasma (**b**) were measured in 55 childhood allergic asthma and 50 healthy controls using RT-qPCR and ELISA, respectively. **c** RT-qPCR showing elevated levels of *STX17* in monocyte-derived macrophages from 55 childhood asthma population. The detailed minimum, median, maximum, 25th, 75th percentile (box), and 5th and 95th percentile (whiskers) of box plots were provided in the Source data file. **d** ROC curve analysis of PTX3 protein levels in 55 childhood asthma. **e** Spearman correlation (two-sided) analysis between PTX3 protein levels in plasma and blood eosinophils numbers, or FeNO in 55 childhood allergic asthma. **f** Proposed working model of m6A-controlled macrophage homeostasis in allergic asthma (Schematic created with BioRender.com). Statistical analysis of the data was performed using two-sided unpaired *t* test (**b**), or Mann–Whitney test (**a**, **c**). Data are presented as means ± SEM. **\**P < 0.01, ****\**\**P < 0.0001.

calculated by normalizing the Ct value of the m⁶A-IP portion to the Ct of the corresponding input portion.

### RNA immunoprecipitation assay

RNA immunoprecipitation (RIP) was performed using the EZ-Magna RIP kit (Millipore) according to the manufacturer's instructions. $1 \times 10^7$ THP1-derived macrophages were lysed with RIP lysis buffer. Cell extracts were co-immunoprecipitated with anti-YTHDF3 (Santa Cruz, sc-377119, 2 μg/ml) or a control IgG antibody, and the retrieved RNA was subjected to RT-qPCR analysis. Normal mouse IgG was used as a negative control.

### Luciferase reporter assay

To generate m⁶A *PTX3* reporters, the psiCheck2 vector (Promega) containing the Renilla and Firefly luciferase genes was used. The wild-type *PTX3* 3'UTR was cloned from THP1-derived macrophages cDNA, while m⁶A-mutant *PTX3* 3'UTR (m⁶A modified by A$^{1333}$ to T substitution) was obtained by DpnI enzyme digestion (Promega). Primer sequences are provided in Supplementary Table 3. For luciferase assays, WT and m⁶A-mutant vectors expressing THP1-derived macrophages were seeded into 96-well plates and were transfected with *METTL3* siRNA, *YTHDF3* siRNA, or negative control (NC) siRNA. After 48 h post-transfection, the Firefly luciferase and Renilla luciferase activity were calculated in each well by a dual-luciferase reporter assay system (Promega). Data were normalized as the value of Renilla luminescence divided by Firefly luminescence.

### mRNAs stability assay

BMDMs from WT or *Mettl3* KO mice, and THP1-derived macrophages with or without *YTHDF3* knock-down, were treated with actinomycin D (Sigma-Aldrich) at a final concentration of 500 ng/ml for 0, 1, 2, or 3 h. Cells were collected and RNA samples were extracted for reverse transcription. The mRNA transcript levels of *PTX3* were detected by RT-qPCR.

### Transmission electron microscopy

Macrophages with *METTL3*, *YTHDF3*, or *PTX3* deficiency were treated with IL-4 and rapamycin (5 μM, MCE) for 12 h. Cells were fixed in 2.5% glutaraldehyde (pH=7.4, 4 °C). After washing, the samples were post-fixed in 1% osmic acid at 4 °C for 2 h and dehydrated in a graded series of ethanol. Subsequently, the samples were infiltrated in propylene oxide (Aladdin, China) and embedded using an SPI-Pon Kit (SPI Science). Ultrathin sections (70 nm) were cut and post-stained with 3% uranyl acetate and 2.7% lead citrate. Samples were visualized using an HT-7800 transmission electron microscope (Hitachi).

### Imaging and analysis of autophagy flux

Tandem fluorescent mRFP-GFP-tagged LC3-adenovirus (HanBio, Shanghai, China) was used to detect the autophagic flux according to the manufacturer's protocol. Macrophages were cultured in Laser confocal plates, and transfected with mRFP-GFP-LC3 adenovirus at 100 multiplicities of infection (MOI) for 24 h. Next, cells were stimulated by IL-4 and rapamycin for 12 h. After washing and fixation, nuclei were stained with DAPI. Image acquisition was performed using a Lecia confocal microscope. Four fields of view were chosen for detection in each group of cells. The number of fluorescent puncta was counted by Image J 1.8 software.

### Statistical analysis

All statistical analyses were performed using GraphPad Prism 9.0 software. Three independent experiments were performed to confirm the reproducibility of each experiment. Bars represent the means ± SEM. Before comparisons between groups, data were tested for Gaussian distribution and homogeneity variance. For data in Gaussian distribution and with homogeneity variance, parametric test was used to analyze, such as independent *t* test, one or two-way ANOVA, etc. For data in non-Gaussian distribution, non-parametric test was used to analyze, such as the Mann–Whitney test, Spearman correlation, etc. For data in Gaussian distribution and without homogeneity variance, Welch's correction was used. *P* values < 0.05 were deemed statistically significant.

### Reporting summary

Further information on research design is available in the Nature Portfolio Reporting Summary linked to this article.

## Data availability

The raw data from the RNA-Seq and m⁶A-Seq have been deposited in the GEO database under accession numbers GSE192533, GSE192726, and GSE193340, respectively. The accessible links of GSE27876 reused in the study are provided in the Supplementary Information. Uncropped and unprocessed scans of blots have been provided as in the Source data file. All other data are available in the article and its Supplementary files or from the corresponding author upon request. Source data are provided with this paper.

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

## Acknowledgements

We thank Ying Wang for carrying out the flow cytometry. This work was supported by grants from the National Key R&D Program of China (2021YFC2701800, 2021YFC2701802 to YZ), National Natural Science Foundation of China (82000033 to LL, 82241038, 81974248 to YZ, 82100033 to SH, 81900751 to XH), Shanghai Rising-Star Program (22QA1401500 to XH), Shanghai Committee of Science and Technology (20ZR1408300 to XH, 21140902400 to YZ, 23ZR1407600, 21ZR1410000 to FJ). The International Joint Laboratory Program of National Children's Medical Center (EK1125180109 to YZ).

## Author contributions

X.H., L. L., S. H., and W. X. conceived and designed the project. X.H., L. L., S. H., W. X., W. Z., C. Z., L. Y., H. Z., Y. G., X. X., Q. L., Z. T., H. Y., J. F., L.W., and X.Z. performed experiments and /or analyzed data. X.H., Y.Z., and L.Q. wrote the manuscript. X.H. and Y.Z. planned, designed, supervised, and coordinated the overall research efforts.

## Competing interests

The authors declare no competing interests.

## Additional information

**Peer review information** : *Nature Communications* thanks Georgina Xanthou, Rebecca Martin and the other, anonymous, reviewer(s) for their contribution to the peer review of this work. A peer review file is available.

