## [Peer Review File · Nature Communications]

RNA m6A Methylation Modulates Airway Inflammation in Allergic Asthma via PTX3-dependent Macrophage HomeostasisREVIEWER COMMENTS

Reviewer #1 Mettl3 (Remarks to the Author):

The manuscript by Han et al presents some lines of evidence for a role of m6A in Allergic Asthma via modulation of macrophage homeostasis. Cre-lox induced knock out of METTL3 in myeloid cells leads to increased inflammatory response in the lungs of a murine allergic asthma model. Consistently METTL3 expression in macrophages negatively correlates with the severity of the disease in children. Lack of METTL3 leads to M2 macrophage activation both in mouse and human cells. RNA seq and meRIP-seq analyses pointed towards PTX3 as a good candidate for m6A target in this process. Indeed KD of PTX3 rescued the inflammatory defects induced by the METTL3 depletion. This function of m6A seems to be mediated through the m6A reader YTHDF3. Finally the authors demonstrated that Ptx3 regulates the expression of STX17 thereby impacting autophagy in macrophages.

This is an impressive amount of work demonstrating the role of m6A in Asthma. Previous reports already found that METTL3 foster M1 activation in macrophages. What is novel here is the identification of the Ptx3 m6A target that inhibits M2 activation. Overall the data are relatively convincing, even though further validation is needed.

Major comments

The m6A-seq data look very noisy and I am not convinced that the experiment indeed worked. First the reported enriched site is different from the consensus one. The authors indicate that GGAC was significantly enriched but it's actually GGAT that is more enriched, which is not normally methylated in other cell types. Second, the authors do not provide the metagene profile where typically m6A is enriched near stop codons. Third the tracks shown in Fig 3G are quite noisy and the enrichment is difficult to detect.

In Fig 3K it's not clear whether the sites that were mutated are indeed methylated. This needs to be validated by an alternative approach (e.g. SCARLET, SELECT).

The result of the YTHDF3 pulldown in Fig 5G is not consistent with the expectation. The depletion of METTL3 should lead to decrease binding of the reader to their targets. How do the authors explain this inconsistency?

The data concerning the PTX3/STX17 axis lacks connection with the m6A pathway. The authors should demonstrate that m6A indeed regulates STX17 expression, as well as autophagy. An interesting experiment would be to test whether the KD of STX17 rescues the decrease of autophagosomes in BMDMs derived from METTL3 KO mice.

Other comments

The quality of the figures is pretty poor. There is a lack of consistency between the fonts of the different panels.

How often the different western blots were performed? Some of them lack quantification.

Some of the conclusions are too affirmative (i.e. lane 266, lanes 293-295).

Reviewer #2 asthma (Remarks to the Author):

NCOMMS-22-37321

In this manuscript, the authors convincingly show that METTL3 expression negatively regulates the development of alternatively activated (M2) macrophages. They perform experiments in mice that genetic deletion of METTL3 using the *Lyz2-Cre* increases allergic airway inflammation after cockroach sensitization and challenge compared to WT mice. They also use the human macrophage cell line THP to show that suppression of METTL3 increases M2 polarization, while THP cells that ectopically express METTL3 preferentially polarized to the M1 phenotype. They further show that phosphorylation of AKT and STAT6 were upregulated in BMDM from METTL3-deficient mice following IL-4 stimulation compared to BMDM from WT mice. A similar result was found with METTL3 knockdown in

THP cells. Further studies suggested that PTX3 was a target for METTL3 and the authors showed that disruption of PTX3 inhibited M2 polarization with genetic depletion of METTL3. Additionally, inhibition of PTX3 reduced the increased airway mucus expression, eosinophilia, and airway responsiveness in the cockroach sensitized and challenged METTL3 KO, seemingly closing the loop on the mechanism. Lastly, loss of METTL3 impaired the YTHDF3-mediated degradation of PTX3 mRNA, showing that YTHDF3 inhibition of PTX3 was critical for METTL3 inhibition of the M2 phenotype.

Major points

There are three major concerns regarding this manuscript, and these are listed individually in the following paragraphs:

First, a major unanswered question is how does the deletion of METTL3 in macrophages change allergen-induced mucus in the airway, airway eosinophils, and airways responsiveness to methacholine? The authors show that STAT6 is upregulated in the macrophages, but how does this regulate an overall increase in Th2 inflammation? In figure 1, the authors do not measure Th2 cytokines in the lungs or BAL fluid. Does the change in macrophage expression of METTL3 have an effect on the recruitment or activation of CD4+ T lymphocytes or the chemokines that recruit Th2 cells to the lung? This could be evaluated by intracellular flow cytometry examining the CD4+ compartment for the production of these cytokines. Also, the BAL and lung homogenates could be assessed for chemokines associated with CD4+ Th2 cell recruitment. How inhibition of METTL3 regulates the cardinal features of asthma in the allergen-challenge model needs to be clearly and comprehensively defined.

Second, in line 136, the title of this section, the authors state "Low METTL3 expression in macrophages from children with allergic asthma is associated with severe disease." This is an extremely misleading statement because it gives the impression that the authors were studying tissue derived macrophages, when they were instead collecting PBMC and then in vitro stimulating the cells to develop a macrophage phenotype. This is very different from what they imply that they are doing in the title. Ideally, the authors would be examining

alveolar macrophages from children because the blood PMBC compartment may not reflect what is transpiring in the lung macrophage. While it is difficult to obtain alveolar macrophages from children with asthma due to the safety concerns associated with bronchoscopy from this population, it does not excuse the authors for being inaccurate in their portrayal of the cells they used in their assays and also to acknowledge the limitation of this approach in the discussion.

The third issue is that the authors did not perform the correct statistical tests. For instance, in line 781 a two-way ANOVA should have been used rather than a one-way ANOVA since there were two factors being assessed - the response to cockroach allergen challenge and the genetics of the mice. More critically, in line 782, the authors state that they used a Pearson correlation, and this is not the correct test as the data certainly does not appear to be normally distributed. Therefore, the authors used have used the nonparametric Spearman correlation, which likely will negate the statistical significance that they report. All the correlation analyses should be performed using the appropriate statistical test and it appears that these correlations may not be significant as they are being driven by a very few values at the extremes of the data set in several of the analyses.

Minor points

Figure 5E- It is not clear why the total STAT6 is changing in this blot. Total cellular STAT6 does not change with increases in p-STAT6.

Reviewer #3 airway macrophages, airway disease (Remarks to the Author):

In the current studies, Xiao Han et. al., observed a decrease in the expression of m6A methyltransferase, METTL3, in peripheral blood macrophages that was associated with the severity of childhood allergic asthma. They further showed that knockout of *Mettl3* in myeloid cells skewed macrophages towards an M2-like phenotype and resulted in exacerbated allergen-driven airway inflammation in vivo. Mechanistic studies demonstrated that loss of METTL3 impaired the m6A-YTHDF3-dependent degradation of PTX3 mRNA, which was associated with enhanced allergic airway inflammation and childhood asthma severity. Finally, a role for PTX3 regulating autophagy maturation in macrophages by

reducing STX17 expression is shown. The findings presented in the study are novel and may have important therapeutic implications pertinent to targeting m6A methyltransferase METTL3 signaling in the context of allergic asthma. However, there are several issues that need to be addressed to strengthen the findings proposed.

Major issues:

1. Crossing *Mettl3^{fl/fl}* mice with *Lyz2-Cre* mice to ablate *Mettl3* in the myeloid compartment does not solely target macrophages/monocytes, but also granulocytes and more specifically neutrophils. The authors need to investigate *Mettl3* expression in airway neutrophils and explore their phenotype in *Mettl3^{-/-}* mice, as depletion of *Mettl3* in neutrophils may also contribute to the observed exacerbated allergic airway disease.
2. The authors should explore Th1, Th17 and Th2 cell-associated cytokine release in the BAL and/or lung homogenates to inform on the effects of *Mettl3* deficiency on the type of the allergic response.
3. What was the phenotype (M1 vs M2) of airway-infiltrating macrophages in the *mettl3^{-/-}* mice?
4. What were the protein levels of *Mettl3* in the peripheral blood macrophages in children with asthma compared to controls?
5. How did the authors define asthma severity? Increased eosinophilic inflammation is not the only marker of disease severity. What about exacerbation frequencies? Which types of medications were these patients on when these measurements were made?
6. Why did the authors use THP-1 monocytes and not peripheral blood macrophages or BMDMs for the *Mettl3* knockdown or overexpression experiments? The authors observations need to be validated in a more physiologically relevant system.
7. What do the authors mean by 'gain of function' experiments? Do they mean overexpression? What are the baseline *Mettl3* levels in these cells?
8. The authors propose that *Mettl3* deficiency skewed macrophages towards an M2-like phenotype through decreasing NF- κ B levels and increasing and activating PI3K/AKT and JAK/STAT6 signaling. This statement is not correct. In order to show this, the authors need to inhibit PI3K/AKT and JAK/STAT6 and/or activate NF κ B and see a reversal of the phenotype.
9. In line 222 (and in other parts of the manuscript) the authors state that *Metll3* deficiency aggravated allergic airway disease phenotype through skewing macrophages towards M2-

like cells. However, they only show association data. To definitely prove this they need to either transfer Metl3^{-/-} airway-infiltrating macrophages in naive mice and see exacerbation of airway inflammation or they need to deplete M2 macrophages in Metl3^{-/-} and see disease amelioration.

10. In line 293, the authors state that 'the upregulation of PTX3 was responsible for the preferential M2 macrophage activation seen in METTL3 deficient macrophages'. To validate this statement, the authors should knockdown PTX3 in Metl3^{-/-} cells and see reversal of the M2 characteristics.

11. Administration of a lentivirus is expected to infect all cells in the lung and not specifically alveolar macrophages. The authors should show that knockdown of PTX3 specifically in macrophages ameliorates disease phenotype in Metl3^{-/-} mice.

12. In figures 1 and 4, better quality histology microphotographs of H&E and PAS staining should be provided.

13. In the immunostaining experiments in Figures 2 and 4, CD206+F4/80+ expressing macrophages are not clearly shown. Higher magnifications should be provided. In Figure 4, PTX3 expressing M2 macrophages are also not clearly shown and higher magnifications should be provided.

Minor issues:

1. It is not clear from the introduction why the authors chose to explore childhood asthma.
2. A concise hypothesis is missing.
3. Certain references are missing (i.e. line 87)
4. The manuscript would benefit from revision by an English native speaker.

Point-by-point responses

To Reviewer: 1

*The manuscript by Han et al presents some lines of evidence for a role of m6A in*
*Allergic Asthma via modulation of macrophage homeostasis. Cre-lox induced knock*
*out of METTL3 in myeloid cells leads to increased inflammatory response in the lungs*
*of a murine allergic asthma model. Consistently METTL3 expression in macrophages*
*negatively correlates with the severity of the disease in children. Lack of METTL3 leads*
*to M2 macrophage activation both in mouse and human cells. RNA seq and meRIP-seq*
*analyses pointed towards PTX3 as a good candidate for m6A target in this process.*
*Indeed KD of PTX3 rescued the inflammatory defects induced by the METTL3 depletion.*
*This function of m6A seems to be mediated through the m6A reader YTHDF3. Finally*
*the authors demonstrated that Ptx3 regulates the expression of STX17 thereby*
*impacting autophagy in macrophages.*
*This is an impressive amount of work demonstrating the role of m6A in Asthma.*
*Previous reports already found that METTL3 foster M1 activation in macrophages.*
*What is novel here is the identification of the Ptx3 m6A target that inhibits M2*
*activation. Overall the data are relatively convincing, even though further validation is*
*needed.*

Response:

We thank the reviewer for carefully reading our manuscript and appreciate the helpful
comments and critical questions. We have studied these issues carefully and provided
our responses point to point as listed below. And based on these suggestions and
questions, we have made changes to the original manuscript to improve our manuscript.
We hope that the revised manuscript would be better for the readers to understand our
points and finally meet with your approval to get published.

.
*1) The m6A-seq data look very noisy and I am not convinced that the experiment indeed*
*worked. First the reported enriched site is different from the consensus one. The authors*

*indicate that GGAC was significantly enriched but it's actually GGAT that is more*
*enriched, which is not normally methylated in other cell types. Second, the authors do*
*not provide the metagene profile where typically m⁶A is enriched near stop codons.*
*Third the tracks shown in Fig 3G are quite noisy and the enrichment is difficult to detect.*

Response:

We are sorry for the unclear description of the m⁶A-seq data analysis, and greatly
appreciate the reviewer's comments. The previously enriched m⁶A motif "GGAC" in
both control and *METTL3*-deficient cells was identified using DREME software,
whereas the significance of motifs analyzed by this software was not very high. Recent
transcriptome-wide m⁶A mapping approaches suggest the consensus m⁶A motif is
"RRACH" (R= G or A; H = A, C or U) ^{1,2}. To carefully identify the sequence features
of m⁶A, we re-performed motif search of m⁶A-enriched regions by MEME software,
and the previously reported consensus "RRACH" motif was identified in both control
and *METTL3*-deficient cells. The statistical significance of these motifs calculated by
MEME showed a lower E-value than DREME, which represented more credibility
(e=4.3e-124 and e=1.4e-083, respectively) (Figure R1A). Meanwhile, consistent with
previous observations ^{2,3}, we found that the density of m⁶A peaks was markedly
enriched near stop codons (Figure R1B). Furthermore, we have marked out the different
m⁶A peak region of *PTX3* transcripts (yellow column) in the original Figure 3G
(renewed as Figure 4G), showing that the *PTX3* m⁶A peaks were markedly enriched in
control cells compared to the *METTL3*-deficient cells (red line) (Figure R1C). We have
amended and added these data to the revised Figure 4.

**Figure R1. Identification of METTL3 targets in macrophages through MeRIP-seq.**

(A) The predominant consensus m⁶A motif RRACH was detected in both control and
 *METTL3*-deficient macrophages through m⁶A-seq. (B) Density distribution of m⁶A
 peaks across mRNA transcripts. (C) Integrative Genomics Viewer (IGV) showing the
 m⁶A abundance on *PTX3* mRNA transcripts in *METTL3*-deficient and control
 macrophages as detected by m⁶A-seq.

*2) In Fig 3K it's not clear whether the sites that were mutated are indeed methylated.*
 *This needs to be validated by an alternative approach (e.g. SCARLET, SELECT).*

Response:

We greatly appreciate the reviewer's valuable suggestion to validate the m⁶A sites of
 *PTX3* using SELECT or SCARLET method ^{4,5}, which would reinforce the conclusion
 that *METTL3* enhances the m⁶A modification of the 3'UTR of *PTX3*. Firstly, by
 analyzing our m⁶A-seq data, we found that *PTX3* harbored two m⁶A sites in its 3' UTR
 (AGA¹³³³CT and AGA¹⁴⁰⁶CA). Subsequently, the SELECT analysis demonstrated that
 the relative amount of SELECT qPCR products targeting the AGA¹³³³CT site was
 significantly enhanced in the *METTL3*-deficient macrophages compared to the control,

whereas no significant difference was observed in the amplification of the AGA¹⁴⁰⁶CA
 site, implying the AGA¹³³³CT site could be methylated by METTL3 (Figure R2A).
 Furthermore, we re-performed luciferase reporter assays with the *PTX3* 3'UTR
 constructs containing wild-type or mutant AGA¹³³³CT sites (m⁶A¹³³³ was replaced by
 74 T). The results showed that *METTL3* or *YTHDF3* deficiency in THP1-derived
 macrophages significantly enhanced the luciferase activity of the reporter construct
 carrying the AGA¹³³³CT site of *PTX3* 3'UTR. This increase was abrogated when the
 putative m⁶A sites were mutated (Figure R2B and R2C), suggesting that the
 downregulation of *PTX3* expression by METTL3 or YTHDF3 was dependent on m⁶A
 modification of the AGA¹³³³CT site in its 3' UTR. We have renewed these data to the
 revised Figure 4, Figure 6, and Supplementary information, respectively.

 **Figure R2. The AGA¹³³³CT site of *PTX3* 3'UTR is methylated by METTL3.** (A)
 The relative amount of SELECT qPCR products targeting the AGA¹³³³CT and
 AGA¹⁴⁰⁶CA site on *PTX3* 3'UTR using the total RNA of THP1-derived macrophages

transfected with *METTL3* or control siRNA. (B) Luciferase reporter and mutagenesis
assays. WT or mutant *PTX3*-3'UTR vector (m⁶A¹³³³ was replaced by T) -transfected
THP1-derived macrophages were treated with *METTL3* or control siRNA. (C)
Luciferase reporter and mutagenesis assays in *YTHDF3*-deficient THP1-derived
macrophages. Data are presented as means ± SEM from three independent experiments.
**P* < 0.05, ****P* < 0.001; n.s = not significant.

*3) The result of the YTHDF3 pulldown in Fig 5G is not consistent with the expectation.*
*The depletion of METTL3 should lead to decrease binding of the reader to their targets.*
*How do the authors explain this inconsistency?*

Response:

We apologized for the carelessness in the Figure preparation. Our m⁶A-seq showed that
the *PTX3* m⁶A level was markedly enriched in control cells compared to the *METTL3*-
deficient cells, whereas *PTX3* mRNA level increased substantially following *METTL3*
depletion, implying the up-regulated *PTX3* transcripts carried hypo-methylated m⁶A
peaks (Hypo-up gene) in *METTL3*-deficient macrophages.

Here, based on the m⁶A AGA¹³³³CT site in *PTX3* transcripts, we re-performed MeRIP
assays and confirmed that the enrichment m⁶A level of *PTX3* transcripts was
significantly down-regulated in *METTL3*-deficient macrophages compared to the
control cells (Figure R3A). Meanwhile, we also re-performed RIP-qPCR assays
targeting the m⁶A AGA¹³³³CT site in *PTX3* transcripts. As expected, *PTX3* m⁶A
transcripts were remarkably enriched in YTHDF3-pull precipitates, while this relative
enrichment was significantly decreased in *METTL3*-deficient macrophages (Figure
R3B). These findings indicated that the METTL3/YTHDF3 axis modulated *PTX3* m⁶A
modification. We have renewed these data to the revised Figure 4 and Figure 6,
respectively.

**Figure R3. METTL3/YTHDF3 axis modulates *PTX3* m⁶A modification.** (A)
 MeRIP-qPCR assay validated the enrichment of m⁶A-modified *PTX3* transcripts in
 *METTL3*-deficient THP1-derived macrophages. The related enrichment of m⁶A in each
 sample was calculated by normalizing the Ct value of the m⁶A-IP portion to the Ct of
 the corresponding input portion. (B) YTHDF3 RIP-analysis of *PTX3* m⁶A level in
 control or *METTL3*-deficient THP1-derived macrophages. Relative enrichment was
 normalized by input. Data are presented as means \pm SEM from three independent
 experiments. *** $P < 0.001$.

*4) The data concerning the PTX3/STX17 axis lacks connection with the m⁶A pathway.*
 *The authors should demonstrate that m⁶A indeed regulates STX17 expression, as well*
 *as autophagy. An interesting experiment would be to test whether the KD of STX17*
 *rescues the decrease of autophagosomes in BMDMs derived from METTL3 KO mice.*

Response:

Thank you for the valuable suggestion to confirm METTL3/YTHDF3 m⁶A axis
 regulates STX17 expression, as well as autophagy, which would strengthen the
 connection between PTX3/STX17 axis and the m⁶A pathway. To address this question,
 firstly, we measured the *STX17* mRNA and protein levels in both *METTL3* and
 *YTHDF3*-deficient macrophages. As expected, the *STX17* mRNA and protein levels
 were largely enhanced in *Mettl3* KO BMDMs, as well as in *YTHDF3* knockdown
 THP1-derived macrophages (Figure R4A and R4B). Subsequently, the data confirmed

that stimulation by rapamycin and IL-4 led to a significant decrease of autophagosomes
in both *Mettl3* KO BMDMs and *YTHDF3* knockdown THP1-derived macrophages
(Figure R4C and R4D). We then transduced mRFP-GFP-LC3 lentivirus into *Mettl3* KO
BMDMs and *YTHDF3* knockdown THP1-derived macrophages, respectively. As
compared with the control cells, the majority of LC3 dots in both *Mettl3* KO BMDMs
and *YTHDF3* knockdown THP1-derived macrophages with autophagy induction just
remained RFP-positive, indicating the marked degradation of autophagosomes in these
cells (Figure R4E). Lastly, to further explore whether the depletion of *Stx17* rescues the
decrease of autophagosomes in *Mettl3* KO BMDMs, we performed TEM assay. The
data showed that the decreased autophagosomes seen in *Mettl3* KO BMDMs could be
largely reversed by the knockdown of *Stx17* (Figure R4F). Collectively, these findings
suggest that METTL3/YTHDF3-m⁶A/PTX3/STX17 axis plays an important role in the
autophagy maturation of macrophages. We have added these data to the revised Figure
7 and Supplementary information.

**Figure R4. METTL3/YTHDF3 axis controls autophagy maturation in**
 **macrophages via an STX17-dependent manner.** (A) RT-qPCR and (B) Western blot
 showing up-regulated *STX17* expression in *Mettl3* KO BMDMs and *YTHDF3*-deficient
 THP1-derived macrophages, respectively. Transmission electron microscopy (TEM)
 demonstrating the decreased autophagosomes in *Mettl3* KO BMDMs (C) and *YTHDF3*-

deficient THP1-derived macrophages (D). Scale bars, 2 μm , and 1 μm , respectively. (E)
The autophagy flux analysis showing the number of LC3 puncta in *Mettl3* KO BMDMs
and *YTHDF3*-deficient THP1-derived macrophages. Scale bars, 25 μm , and 10 μm ,
respectively. (F) Analysis of the autophagosome number in *Mettl3* KO BMDMs with
or without *siStx17* knockdown. Scale bars, 1 μm . Data are presented as means \pm SEM
from three independent experiments. * $P < 0.05$, ** $P < 0.01$, *** $P < 0.001$.

*5) The quality of the figures is pretty poor. There is a lack of consistency between the*
*fonts of the different panels.*

Response:

We are sorry for the poor quality of the figures and the lack of consistency between the
fonts of the different panels. We have amended them in the revised manuscript.

*6) How often the different western blots were performed? Some of them lack*
*quantification.*

Response:

We appreciate the reviewer's reminder. We added the following description in the
Statistical analysis: "For Western blot, representative figures from three biological
replicates were shown". Meanwhile, we have quantified all the western blots in the
revised manuscript.

*7) Some of the conclusions are too affirmative (i.e. lane 266, lanes 293-295).*

Response:

Thanks for the reviewer's suggestion. As the reviewer commented, we have modified
and tuned down the statement in the revised manuscript as follows:

"indicating that *PTX3* is a target of METTL3".

"Similar observations were made in THP1-derived macrophages after *PTX3*
knockdown (Supplementary Fig. 9), implying that the downregulation of *PTX3*
inhibited M2 macrophage activation".

To Reviewer: 2

*In this manuscript, the authors convincingly show that METTL3 expression negatively*
*regulates the development of alternatively activated (M2) macrophages. They perform*
*experiments in mice that genetic deletion of METTL3 using the Lyz2-Cre increases*
*allergic airway inflammation after cockroach sensitization and challenge compared to*
*WT mice. They also use the human macrophage cell line THP to show that suppression*
*of METTL3 increases M2 polarization, while THP cells that ectopically express*
*METTL3 preferentially polarized to the M1 phenotype. They further show that*
*phosphorylation of AKT and STAT6 were upregulated in BMDM from METTL3-*
*deficient mice following IL-4 stimulation compared to BMDM from WT mice. A similar*
*result was found with METTL3 knockdown in THP cells. Further studies suggested that*
*PTX3 was a target for METTL3 and the authors showed that disruption of PTX3*
*inhibited M2 polarization with genetic depletion of METTL3. Additionally, inhibition*
*of PTX3 reduced the increased airway mucus expression, eosinophilia, and airway*
*responsiveness in the cockroach sensitized and challenged METTL3 KO, seemingly*
*closing the loop on the mechanism. Lastly, loss of METTL3 impaired the YTHDF3-*
*mediated degradation of PTX3 mRNA, showing that YTHDF3 inhibition of PTX3 was*
*critical for METTL3 inhibition of the M2 phenotype.*

Response:

We thank the reviewer for carefully reading our manuscript and appreciate the helpful
comments and critical questions. We have studied these issues carefully and provided
our responses point to point as listed below. And based on these suggestions and
questions, we have made changes to the original manuscript to improve our manuscript.
We hope that the revised manuscript would be better for the readers to understand our
points and finally meet with your approval to get published.

*1) First, a major unanswered question is how does the deletion of METTL3 in*
*macrophages change allergen-induced mucus in the airway, airway eosinophils, and*

*airways responsiveness to methacholine? The authors show that STAT6 is upregulated*
*in the macrophages, but how does this regulate an overall increase in Th2 inflammation?*
*In figure 1, the authors do not measure Th2 cytokines in the lungs or BAL fluid. Does*
*the change in macrophage expression of METTL3 have an effect on the recruitment or*
*activation of CD4+ T lymphocytes or the chemokines that recruit Th2 cells to the lung?*
*This could be evaluated by intracellular flow cytometry examining the CD4+*
*compartment for the production of these cytokines. Also, the BAL and lung*
*homogenates could be assessed for chemokines associated with CD4+ Th2 cell*
*recruitment. How inhibition of METTL3 regulates the cardinal features of asthma in*
*the allergen-challenge model needs to be clearly and comprehensively defined.*

**Response:**

We greatly appreciate the reviewer's suggestion to measure Th2 cytokines in the lungs
in vivo mouse models, which would reinforce the conclusion that METTL3 plays a key
role in allergic asthma development via M2 macrophage activation.

Allergic asthma has been generally considered as a Th2 cell-mediated chronic immune
response, although Th1 cell/Th17 cell immunity may involve certain aspects of this
disease ^{6, 7}. Th2 cells produce effector cytokines such as IL-4, IL-5, and IL-13 to
mediate respiratory symptoms, correlating with the degree of airway eosinophilia. More
importantly, recent studies reveal that M2 macrophage activation plays a crucial role in
allergic asthma through expressing high levels of chemokines, including CCL-17,
CCL-22, and CCL-24 ⁸. The release of these cytokines results in the recruitment of Th2
cells and amplification of polarized Th2 responses, leading to the infiltration of
eosinophil infiltration into the bronchial tissues ^{9, 10}. To further determine whether
*METTL3* depletion promotes Th2 responses in allergic inflammation via M2
macrophage activation, the levels of M2-associated chemokines (i.e., *Ccl-17*, and *Ccl-*
*22*) in alveolar macrophages were tested. We found that the mRNA levels of *Ccl-17* and
*Ccl-22* were significantly elevated in alveolar macrophages from CRE-challenged
*Mettl3* KO mice, compared to CRE-challenged WT animals (Figure R5A). Furthermore,
compared to WT mice, the mRNA and protein levels of Th2 cell-associated cytokines

(i.e., *IL-4*, and *IL-13*) were markedly increased in lung homogenates from CRE-
 challenged *Mettl3* KO mice, whereas the Th1 cell-associated cytokine (*ifng*) showed
 downregulated expression and the Th17 cell-associated cytokine (*Il-17a*) had no
 difference (Figure R5B and R5C). Lastly, we demonstrated that IL-4-producing Th2
 cells were also enhanced in the Mediastinal lymph nodes (MLNs) from CRE-
 challenged *Mettl3* KO mice, while IFN-gamma-producing Th1 cells were
 comparatively reduced (Figure R5D). Thus, the above findings suggest that *Mettl3*
 deficiency promotes Th2 responses and accelerates airway inflammation in allergic
 asthma via M2 macrophage activation. We have added these data to the revised Figure
 3.

**Figure R5. *Mettl3* deficiency promotes Th2 responses in allergic asthma via M2**

**macrophage activation.** (A) RT-qPCR showing up-regulated M2-associated
chemokines in alveolar macrophages purified from CRE allergen-induced asthma
models. The levels of Th1, Th2, and Th17 cell-associated cytokines in lung
homogenates were detected by RT-qPCR (B) and ELISA (C), respectively. (D) Flow
cytometry analysis of the frequency of CD4⁺IL-4⁺Th2 cells and CD4⁺IFN-gamma⁺Th1
cells in MLNs from mice. Data are presented as means ± SEM and representative of
two independent experiments. **P* < 0.05, ***P* < 0.01, ****P* < 0.001; n.s = not significant.

*2) Second, in line 136, the title of this section, the authors state "Low METTL3*
*expression in macrophages from children with allergic asthma is associated with*
*severe disease." This is an extremely misleading statement because it gives the*
*impression that the authors were studying tissue derived macrophages, when they were*
*instead collecting PBMC and then in vitro stimulating the cells to develop a*
*macrophage phenotype. This is very different from what they imply that there are doing*
*in the title. Ideally, the authors would be examining alveolar macrophages from*
*children because the blood PMBC compartment may not reflect what is transpiring in*
*the lung macrophage. While it is difficult to obtain alveolar macrophages from children*
*with asthma due to the safety concerns associated with bronchoscopy from this*
*population, it does not excuse the authors for being inaccurate in their portrayal of the*
*cells they used in their assays and also to acknowledge the limitation of this approach*
*in the discussion.*

Response:

We are sorry for the inaccurate description in the Results. We have now amended it as
follows: "Low *METTL3* expression in monocyte-derived macrophages from children
with allergic asthma is associated with disease severity". Meanwhile, to exclude any
misunderstanding, we corrected "macrophages" to "monocyte-derived macrophages"
in the part of clinical samples analysis.

Since it is difficult to obtain alveolar macrophages from childhood asthma due to safety
concerns, we further added the limitation of this approach in the revised discussion as

follows: “In addition, although our studies with monocyte-derived macrophages in
children with allergic asthma and myeloid cells in animal models have suggested a
crucial role for METTL3/m⁶A modification in the development of allergic asthma
mediated by macrophage activation, lung tissue-resident macrophages evidence for this
phenotype in childhood asthma is lacking, warranting further studies”.

*3) The third issue is that the authors did not perform the correct statistical tests. For*
*instance, in line 781 a two-way ANOVA should have been used rather than a one-way*
*ANOVA since there were two factors being assessed - the response to cockroach*
*allergen challenge and the genetics of the mice. More critically, in line 782, the authors*
*state that they used a Pearson correlation, and this is not the correct test as the data*
*certainly does not appear to be normally distributed. Therefore, the authors used have*
*used the nonparametric Spearman correlation, which likely will negate the statistical*
*significance that they report. All the correlation analyses should be performed using*
*the appropriate statistical test and it appears that these correlations may not be*
*significant as they are being driven by a very few values at the extremes of the data set*
*in several of the analyses.*

Response:

We are really sorry for our careless mistakes in statistical analysis. Here, we re-
performed data analysis between multiple groups by the two-way ANOVA with the
post-hoc Bonferroni test. The results showed that this analysis didn't change our
original conclusion. We have renewed these data in the revised manuscript.

Meanwhile, we sincerely appreciate the reviewer's comments on the correlation
analysis. We carefully analyzed the data again using Spearman correlation, and found
that some asthma patients had the extremes levels of eosinophils number and FeNO,
leading to the clear outliers. Although the outliers straightforwardly support our
correlation analysis, it may mislead the readers that the statistical significance we got
here is attributed to these outliers. Thus, to make the result more convincible, we
excluded these outliers, and recruited another fifteen new children with allergic asthma

and ten healthy controls. After the addition of these samples and excluding the previous
outliers (total of 50 normal controls and 50 asthma patients), we re-performed the
Spearman correlation analysis. Consistent with our previous conclusion, the results
suggested that the expression of *METTL3* in monocyte-derived macrophages from
children with allergic asthma was negatively correlated with disease severity, while the
PTX3 circulating levels showed a positive correlation with asthma severity (Figure R6).
Thus, in the revised manuscript, we used Figure R6 instead of original data to make
reader clear.

**Figure R6. Clinical correlation between METTL3, PTX3 and disease severity in**
 **childhood allergic asthma. (A) The transcripts levels of *METTL3* in PBMCs and**
 **monocyte-derived macrophages from 50 childhood allergic asthma and 50 healthy**
 **controls, respectively. (B) ROC curve analysis of monocyte-derived macrophages**
 ***METTL3* levels in childhood asthma. (C) Spearman correlation analysis of monocyte-**

derived macrophages *METTL3* expression, blood eosinophils number, FeNO,
 and %FEV₁ levels in childhood asthma. (D) Elevated levels of *PTX3* and *STX17* in
 monocyte-derived macrophages from childhood asthma relative to healthy subjects. (E)
 The upregulated protein levels of *PTX3* in plasma from childhood asthma and (F) ROC
 curve analysis. (G) Spearman correlation analysis between *PTX3* protein levels and
 blood eosinophils numbers, or FeNO in childhood asthma. **P* < 0.05, ***P* < 0.01, ****P*
 < 0.001.

4) Figure 5E- It is not clear why the total *STAT6* is changing in this blot. Total cellular
 *STAT6* does not change with increases in *p-STAT6*.

Response:

We apologized for the carelessness in the Figure preparation. We re-performed the
 western bolt assay and amended it as follows (Figure R7):

**Figure R7. Elevated levels of p-AKT and p-STAT6 in *YTHDF3*-deficient THP1-**
 **derived macrophages were detected.** **P* < 0.05, ***P* < 0.01; n.s = not significant.

To Reviewer: 3

*In the current studies, Xiao Han et. al., observed a decrease in the expression of m6A*
 *methyltransferase, METTL3, in peripheral blood macrophages that was associated*
 *with the severity of childhood allergic asthma. They further showed that knockout of*

*Mettl3* in myeloid cells skewed macrophages towards an M2-like phenotype and resulted
in exacerbated allergen-driven airway inflammation in vivo. Mechanistic studies
demonstrated that loss of *METTL3* impaired the m6A-YTHDF3-dependent degradation
of *PTX3* mRNA, which was associated with enhanced allergic airway inflammation and
childhood asthma severity. Finally, a role for *PTX3* regulating autophagy maturation
in macrophages by reducing *STX17* expression is shown. The findings presented in the
study are novel and may have important therapeutic implications pertinent to targeting
m6A methyltransferase *METTL3* signaling in the context of allergic asthma. However,
there are several issues that need to be addressed to strengthen the findings proposed.

Response:

We thank the reviewer for carefully reading our manuscript and appreciate the helpful
comments and critical questions. We have studied these issues carefully and provided
our responses point to point as listed below. And based on these suggestions and
questions, we have made changes to the original manuscript to improve our manuscript.
We hope that the revised manuscript would be better for the readers to understand our
points and finally meet with your approval to get published.

*1) Crossing *Mettl3*^{fl/fl} mice with *Lyz2*-Cre mice to ablate *Mettl3* in the myeloid
compartment does not solely target macrophages/monocytes, but also granulocytes and
more specifically neutrophils. The authors need to investigate *Mettl3* expression in
airway neutrophils and explore their phenotype in *Mettl3*^{-/-} mice, as depletion of *Mettl3*
in neutrophils may also contribute to the observed exacerbated allergic airway disease.*

Response:

We sincerely appreciate the reviewer's reminder. Although allergic asthma is classically
associated with eosinophilia and Th2 cytokines, neutrophils may involve certain aspects
of this disease ⁶. To address this question, we first purified neutrophils from the bone
marrow of the experimental mice using the EasySep Mouse Neutrophil Enrichment Kit
(STEMCELL). Neutrophils from the *Lyz2*-Cre conditional *Mettl3* KO mice showed a
reduction in *METTL3* protein levels (Figure R8A). Compared to WT mice, no apparent

abnormalities of neutrophils from the bone marrow were noted in *Mettl3* KO mice
(Figure R8B). Next, in the CRE-challenged allergic asthma model, we detected Gr1
expression in lung tissues through immunohistochemistry (IHC). Much higher numbers
of Gr1⁺ neutrophils were detected after CRE challenge as compared with PBS-treated
mice. Noticeably, compared with CRE-treated WT mice, the infiltration of Gr1⁺
neutrophils was enhanced in CRE-treated *Mettl3* KO mice (Figure R8C). To further
examine whether the accelerated airway inflammation in *Mettl3* KO mice was related
to the neutrophils, we depleted the neutrophils in CRE-induced asthma models by i.p
injecting anti-Ly6G Ab or isotype control Ab ¹¹. Flow cytometry analysis confirmed a
marked reduction in the percentages of neutrophil infiltration in BALF from mice
treated with anti-Ly6G Ab (Figure R8D). Meanwhile, we found that the depletion of
neutrophils by anti-Ly6G Ab could not abrogate the increased airway inflammation
phenotype of *Mettl3* KO mice (Figure R8E-G). The above findings indicated that the
function of METTL3 in allergic airway inflammation was not dependent on neutrophils.
We have added these data to the revised Supplementary information.

**Figure R8. Depletion of neutrophils does not reduce the differences of airway**

**inflammation between *Mettl3* KO and WT mice.** (A) Western blot showing reduced

METTL3 protein levels in neutrophils purified from the bone marrow of the

experimental mice. (B) The percentage of neutrophils from the bone marrow was

detected in *Mettl3* KO and WT mice by flow cytometry. (C) Representative images of

Gr1 expression in lung tissues using IHC. Scale bars: 200 μ m. Every 72 h during CRE
treatment, mice were i.p. injected with 200 μ g anti-Ly6G mAb or isotype control mAb,
(D) Flow cytometry analysis of the efficiency of neutrophils depletion in BALF. (E)
Total and differential BALF cell numbers, and (F) histopathological changes in the lung
tissues were examined. Scale bars: 200 μ m and 100 μ m, respectively. (G) Calculated
inflammation and PAS scores. Data are presented as means \pm SEM and representative
of two independent experiments. * P < 0.05, ** P < 0.01, *** P < 0.001; n.s = not
significant.

*2) The authors should explore Th1, Th17 and Th2 cell-associated cytokine release in*
*the BAL and/or lung homogenates to inform on the effects of Mettl3 deficiency on the*
*type of the allergic response.*

Response:

Thank the reviewer for the valuable suggestion. Allergic asthma has been generally
considered as a Th2 cell-mediated chronic immune response, although Th1 cell/Th17
cell immunity may involve certain aspects of this disease^{6,7}. Th2 cells produce effector
cytokines such as IL-4, IL-5, and IL-13 to mediate respiratory symptoms, correlating
with the degree of airway eosinophilia. More importantly, recent studies reveal that M2
macrophage activation plays a crucial role in allergic asthma through expressing high
levels of chemokines, including CCL-17, CCL-22, and CCL-24⁸. The release of these
cytokines results in the recruitment of Th2 cells and amplification of polarized Th2
responses, leading to the infiltration of eosinophil infiltration into the bronchial tissues
420^{9,10}. To further determine whether *Mettl3* depletion promotes Th2 responses in allergic
inflammation via M2 macrophage activation, the levels of M2-associated chemokines
(i.e., *Ccl-17*, and *Ccl-22*) in alveolar macrophages were tested. We found that the
mRNA levels of *Ccl-17* and *Ccl-22* were significantly elevated in alveolar macrophages
from CRE-challenged *Mettl3* KO mice compared to CRE-challenged WT animals
(Figure R5A). Furthermore, compared to WT mice, the mRNA and protein levels of
Th2 cell-associated cytokines (i.e., *IL-4*, and *IL-13*) were markedly increased in lung

homogenates from CRE-challenged *Mettl3* KO mice, whereas the Th1 cell-associated
 cytokine (*ifng*) showed downregulated expression and the Th17 cell-associated
 cytokine (*Il-17a*) had no difference (Figure R5B and R5C). Lastly, we demonstrated
 that IL-4-producing Th2 cells were also enhanced in the Mediastinal lymph nodes
 (MLNs) from CRE-challenged *Mettl3* KO mice, while IFN-gamma-producing Th1
 cells were comparatively reduced (Figure R5D). Thus, the above findings suggest that
 *Mettl3* deficiency promotes Th2 responses and accelerates airway inflammation in
 allergic asthma via M2 macrophage activation. We have added these data to the revised
 Figure 3.

 **Figure R5. *Mettl3* deficiency promotes Th2 responses in allergic airway**
 **inflammation via M2 macrophage activation.** (A) RT-qPCR showing up-regulated
 M2-associated chemokines in alveolar macrophages purified from CRE allergen-

induced asthma models. (B) (C) The levels of Th1, Th2, and Th17 cell-associated
cytokines in lung homogenates were detected by RT-qPCR and ELISA, respectively.
(D) Flow cytometry analysis of the frequency of CD4⁺IL-4⁺Th2 cells and CD4⁺IFN-
gamma⁺Th1 cells in MLNs from mice. Data are presented as means ± SEM and
representative of two independent experiments. **P* < 0.05, ***P* < 0.01, ****P* < 0.001;
n.s = not significant.

*3) What was the phenotype (M1 vs M2) of airway-infiltrating macrophages in the*
*mettl3^{-/-} mice?*

Response:

Thank you for the valuable suggestion. Alveolar macrophages are critical resident cells
in the alveolus, which are important for both lung homeostasis and the response to
injury. Recently studies have confirmed that there are two ontologically distinct
populations of alveolar macrophages. Tissue-resident alveolar macrophages (TR-AMs)
differentiate shortly after birth and persist over the lifespan via self-renewal. Monocyte-
derived alveolar macrophages (Mo-AMs) develop from circulating monocytes and are
recruited to the lung during injury ¹². Generally, alveolar macrophages are identified
based on the expression of specific surface markers such as CD11c, CD64, F4/80,
MerTK, and Siglec F. Furthermore, differences in the levels of expression of Siglec F
allow for discrimination of Mo-AMs (CD11C⁺SiglecF^{int/low}) and TR-AMs
(CD11C⁺SiglecF^{high}) during the course of lung injury ¹³. In our study, flow cytometry
analysis confirmed that compared with CRE-treated WT mice, the percentage of M2
macrophages (CD206⁺CD86⁻) in Mo-AMs from BALF was higher in CRE-treated
*Mettl3* KO mice, while the percentage of M1 macrophages (CD86⁺CD206⁻) decreased
(Figure R9). These data imply that *Mettl3* deficiency promotes M2 macrophage
activation in Mo-AMs during airway inflammation. We hope this evidence will be
helpful.

**Figure R9. *Mettl3* deficiency promotes M2 macrophage activation in Mo-AMs**

**during airway inflammation.** (A) Representative flow cytometry plots gated on

CD45⁺ living cells isolated from BALF from CRE-treated WT and *Mettl3* KO mice.

CD11c^{high}SiglecF^{int/low} Mo-AMs; CD11c^{high}SiglecF^{high} TR-AMs; CD86⁺CD206⁻ M1

macrophages; CD206⁺CD86⁻ M2 macrophages. (B) The percentage of M1 and M2

macrophages in Mo-AMs was quantified. Data are presented as means \pm SEM and

representative of two independent experiments. ** $P < 0.01$, *** $P < 0.001$.

*4) What were the protein levels of Mettl3 in the peripheral blood macrophages in*

*children with asthma compared to controls?*

Response:

We sincerely appreciate the reviewer's reminder. Western blot analysis verified that the

protein levels of METTL3 in monocyte-derived macrophages from children with

allergic asthma were markedly reduced, as compared with normal controls (Figure R10).

We have added these data to the revised Supplementary information.

**Figure R10. The Lower METTL3 levels in monocyte-derived macrophages from**
 **children with allergic asthma.** The METTL3 protein levels were determined in
 monocyte-derived macrophages from children with allergic asthma and healthy
 controls by Western blot. * $P < 0.05$.

*5) How did the authors define asthma severity? Increased eosinophilic inflammation is*
 *not the only marker of disease severity. What about exacerbation frequencies? Which*
 *types of medications were these patients on when these measurements were made?*

Response:

Thanks for the reviewer's suggestion. The current concept of asthma severity,
 recommended by GINA and most asthma guidelines, is that it should be assessed
 retrospectively from how difficult the patient's asthma is to treat ^{14, 15}. In our study,
 asthma severity was estimated using the medication use information reported in
 outpatient pharmacy records according to step-treatment recommendations by the
 GINA criteria, assessment of asthma control using the Childhood Asthma Control Test
 (C-ACT), frequency of asthma exacerbations, and lung function. Here, we defined that
 mild asthma was well controlled with low-intensity treatment, i.e., as-needed low-dose
 ICS-formoterol, or low-dose ICS plus as-needed SABA, while moderate asthma was
 defined as asthma that was well controlled with Step 3 or Step 4 treatment e.g. with
 low- or medium-dose ICS-LABA in either treatment track. Meanwhile, we defined
 severe asthma that remained uncontrolled despite optimized treatment with high-dose
 ICS-LABA, or that required high-dose ICS-LABA to prevent it from becoming
 uncontrolled.

However, reliance on the type/dose of prescribed medication and symptom control does
not adequately capture those at risk of adverse outcomes, suggesting the importance of
biomarkers for risk and treatment stratification ¹⁶. Noticeably, accumulating studies
have demonstrated that eosinophilic inflammation is frequently associated with
increased asthma severity, while the use of peripheral blood eosinophil counts as a
biomarker for increased disease burden or exacerbation risk is more attractive and
feasible ^{16, 17, 18}. Most patients with allergic asthma have predominant type 2
inflammation-mediated disease, and eosinophilic inflammation appears to be close to
the risk of asthma exacerbations and loss of asthma control with inhaled corticosteroid
withdrawal ^{16, 17}. Thus, in our study, a composite type-2 biomarker of blood eosinophils
and FeNO was used to improve the prediction of asthma attacks. We hope this evidence
will be useful.

*6) Why did the authors use THP-1 monocytes and not peripheral blood macrophages*
*or BMDMs for the Mettl3 knockdown or overexpression experiments? The authors*
*observations need to be validated in a more physiologically relevant system.*

Response:

Thanks for the reviewer's comments. In the present study, we detected the effect of
METTL3 on macrophage homeostasis using both BMDMs and THP1-derived
macrophages, which ensured functional conservation between human and mouse
species. Combining with BMDMs from WT and *Mettl3* KO mice, we highlighted the
critical role of m⁶A in regulating macrophage activation. Meanwhile, we also confirmed
that overexpressed *Mettl3* (*Mettl3* LV) promoted M1 and inhibited M2 macrophage
activation in BMDMs (Figure R11). Since the isolation of macrophages from childhood
asthma PBMCs is relatively laborious, to further determine the potential importance of
METTL3 in humans, we used the THP1-derived macrophages to perform the
knockdown or overexpression experiments. It is well-known that PMA is an effective
differentiation agent to obtain mature THP-1 monocyte-derived macrophages with
similarities to PBMC monocyte-derived macrophages ¹⁹. There are notable advantages

in the use of THP1-derived macrophages over PBMC-derived macrophages: easy
 acquisition and handling, unlimited cell number, homogeneous genetic/epigenetic
 backgrounds, and purity of macrophage population²⁰. Thus, in our study, we treated
 the THP1 monocytes with PMA to generate macrophage-like cells, and identified the
 crucial role of m⁶A in regulating macrophage homeostasis in humans. We have added
 these data to the revised Supplementary information.

**Figure R11. Overexpressed *Mettl3* enhances M1 and inhibits M2 macrophage**
 **activation in BMDMs.** Overexpression of *Mettl3* in BMDMs with *Mettl3* LV or Ctrl
 LV. M1 (left)-and M2 (right)-associated markers were quantified by RT-qPCR in
 macrophages stimulated with LPS or IL-4, respectively. Data are presented as means ±
 SEM from three independent experiments. * $P < 0.05$, ** $P < 0.01$, *** $P < 0.001$; n.s =
 not significant.

*7) What do the authors mean by 'gain of function' experiments? Do they mean*
 *overexpression? What are the baseline Mettl3 levels in these cells?*

Response:

Thanks for the reviewer's comments. In our study, the "gain-of-function" studies mean
 the overexpression of *METTL3* in BMDMs and human THP1-derived macrophages. A
 higher level of *Mettl3* relative to β-actin was detected in BMDMs, while *Mettl3* LV
 markedly enhanced the expression of *Mettl3* (Figure R12).

**Figure R12. The levels of *Mettl3* in BMDMs.** In WT mice, RT-qPCR showed the
 levels of *Mettl3* relative to β -actin in BMDMs treated with *Mettl3* LV or controls. *** P
 < 0.001 ; n.s = not significant.

*8) The authors propose that *Mettl3* deficiency skewed macrophages towards an M2-*
 *like phenotype through decreasing NF- κ B levels and increasing and activating*
 *PI3K/AKT and JAK/STAT6 signaling. This statement is not correct. In order to show*
 *this, the authors need to inhibit PI3K/AKT and JAK/STAT6 and/or activate NF κ B and*
 *see a reversal of the phenotype.*

Response:

We sincerely appreciate the reviewer's reminder. To investigate whether the role of
 METTL3 in M2 macrophage activation is dependent on PI3K/AKT and JAK/STAT6
 signaling, we repressed the activation of AKT and STAT6 proteins in BMDMs from
 WT and *Mettl3* KO mice, using the AKT inhibitor, GSK690693, and the STAT6
 inhibitor, AS1517499, respectively ^{21, 22}. The rescue experiments demonstrated that
 *Mettl3* deficiency enhanced M2-associated genes levels, whereas the inhibition of AKT
 or STAT6 phosphorylation levels eliminated this effect (Figure R13), implying the role
 of *METTL3* in M2 macrophage activation dependent on PI3K/AKT and JAK/STAT6
 signaling. We have added these data (Figure R13A) to the revised Supplementary
 information.

**Figure R13. The effect of METTL3 in M2 macrophage activation is dependent on**

**PI3K/AKT and JAK/STAT6 signaling.** (A) In WT and *Mettl3* mice, RT-qPCR

detected M2-associated markers expression in BMDMs treated with the AKT inhibitor

GSK690693 (100 nM), or the STAT6 inhibitor AS1517499 (100 nM). (B) (C) Western

blot showing levels of p-AKT and p-STAT6 in these cells. Data are presented as means

\pm SEM from three independent experiments. * $P < 0.05$, ** $P < 0.01$, *** $P < 0.001$

*9) In line 222 (and in other parts of the manuscript) the authors state that Metll3*

*deficiency aggravated allergic airway disease phenotype through skewing*

*macrophages towards M2-like cells. However, they only show association data. To*

*definitely prove this they need to either transfer Metll3-/- airway-infiltrating*

*macrophages in naive mice and see exacerbation of airway inflammation or they need*

*to deplete M2 macrophages in Metll3-/- and see disease amelioration.*

Response:

We greatly appreciate the reviewer's valuable suggestion. To address this question, the

ideal model is to validate it by transferring *Mettl3* KO mice macrophages in naive mice.

However, due to the limited time, to confirm whether the role of METTL3 in airway

inflammation is due to macrophage-intrinsic effect, we depleted macrophages in vivo
 using clodronate-containing liposomes (CLs)^{11,23}. In the CRE-induced asthma model,
 flow cytometry analysis confirmed a significant decrease in the percentage of alveolar
 macrophages in BALF from *Mettl3* KO mice treated with CLs, compared to controls
 (PBS)-treated *Mettl3* KO mice (Figure R14A). In addition, depletion of macrophages
 completely reversed the susceptibility of *Mettl3* KO mice to allergic airway
 inflammation, compared with wild-type animals (Figure R14B-E), suggesting that the
 vital role of METTL3 in airway inflammation is dependent on macrophages. We have
 added these data to the revised Figure 2.

**Figure R14. The effect of METTL3 on airway inflammation is dependent on**

**macrophages.** Every 72 h during CRE treatment, CLs-liposome (200 μ l) was
administered intratracheally in the clodronate group and PBS was administered in the
control group. (A) Flow cytometry analysis of the efficiency of macrophage depletion
in BALF from *Mettl3* KO mice by clodronate treatment. (B) Total and (C) differential
BALF cell numbers from experimental animals were analyzed by flow cytometry. (D)
Histopathological changes in the lung tissues were examined by H&E- and PAS-
staining. Scale bars: 200 μ m and 100 μ m, respectively. (E) Calculated inflammation
and PAS scores. Data are presented as means \pm SEM and representative of two
independent experiments. * $P < 0.05$, ** $P < 0.01$, *** $P < 0.001$; n.s = not significant.

*10) In line 293, the authors state that 'the upregulation of PTX3 was responsible for the*
*preferential M2 macrophage activation seen in METTL3 deficient macrophages'. To*
*validate this statement, the authors should knockdown PTX3 in Mettl3-/- cells and see*
*reversal of the M2 characteristics.*

Response:

We sincerely appreciate the reviewer's valuable suggestion. Here, the rescue studies
demonstrated that the M2-associated genes levels, the *Ptx3* mRNA levels, and the
phosphorylation levels of AKT and STAT6 proteins were increased in BMDMs from
*Mettl3* KO mice, whereas this enhanced effect of *Mettl3* deficiency on M2 macrophage
activation was markedly abolished by *Ptx3* knockdown (Figure R15), indicating that
the upregulation of PTX3 was responsible for the preferential M2 macrophage
activation seen in *Mettl3*-deficient macrophages. We have added these data (Figure
R15A) to the revised Figure 5.

**Figure R15. The enhanced effect of METTL3 deficiency on M2 macrophage**

**activation is dependent on PTX3.** (A) RT-qPCR (B) and Western blot showing M2

activation-associated markers expression in BMDMs from WT and *Mettl3* KO mice,

with or without *Ptx3* knockdown. Data are presented as means \pm SEM. * $P < 0.05$, *** P

< 0.001 .

*11) Administration of a lentivirus is expected to infect all cells in the lung and not*

*specifically alveolar macrophages. The authors should show that knockdown of PTX3*

*specifically in macrophages ameliorates disease phenotype in *Mettl3*^{-/-} mice.*

Response:

We sincerely appreciate the reviewer's valuable suggestion. To address this question,

the ideal model is to validate it using *Ptx3* /*Mettl3* macrophage-specific knockout mice.

However, due to the limited time and funding, to further elucidate the role of *Ptx3*

knockdown in allergic airway inflammation dependent on macrophages, we depleted

macrophages in vivo using clodronate-containing liposomes (CLs). Similar to our

findings in *Mettl3* KO mice, the percentage of alveolar macrophages was significantly

reduced in BALF from *Ptx3* knockdown mice treated with CLs, compared to controls

(PBS)-treated *Ptx3* knockdown mice (Figure R16A). Furthermore, the data showed that
 compared to shCtrl–infected mice, *Ptx3* knockdown in vivo noticeably alleviated the
 CRE-induced allergic airway inflammation. However, we found that, through the
 depletion of macrophages using CLs, there was no significant difference in airway
 inflammation between sh*Ptx3*–treated mice (sh*Ptx3* CLs) and shCtrl–mice (shCtrl CLs)
 (Figure R16B-E), suggesting that the role of *Ptx3* knockdown in allergic airway
 inflammation may be due to macrophages. We have added these data to the revised
 Supplementary information and discussed the limitation in the discussion section.

**Figure R16. The effect of PTX3 on airway inflammation is dependent on**
 **macrophages.** Every 72 h during CRE treatment, CLs-liposome (200 μl) was
 administered intratracheally in the clodronate group and PBS was administered in the

control group. The sh*Ptx3* lentivirus or shCtrl virus was administered by intratracheal
 instillation on day 14. (A) Flow cytometry analysis of the efficiency of macrophage
 depletion in BALF from sh*Ptx3*-treated mice by clodronate treatment. (B) Total and
 (C) differential BALF cell numbers from experimental animals were analyzed by flow
 cytometry. (D) Histopathological changes in the lung tissues were examined by H&E-
 and PAS-staining. Scale bars: 200 μm and 100 μm , respectively. (E) Calculated
 inflammation and PAS scores. Data are presented as means \pm SEM and representative
 of two independent experiments. * $P < 0.05$, ** $P < 0.01$, *** $P < 0.001$; n.s = not
 significant.

*12) In figures 1 and 4, better quality histology microphotographs of H&E and PAS*
 *staining should be provided.*

Response:

Thanks for the reviewer's reminder. We have provided the higher magnification of
 histology microphotographs in the revised manuscript as follows (Figure R17).

**Figure R17. Representative histology microphotographs in experimental animals.**

Scale bars: H&E staining 200 μ m, and PAS staining 100 μ m, respectively.

*13) In the immunostaining experiments in Figures 2 and 4, CD206+F4/80+ expressing*
*macrophages are not clearly shown. Higher magnifications should be provided. In*
*Figure 4, PTX3 expressing M2 macrophages are also not clearly shown and higher*
*magnifications should be provided.*

Response:

Thanks for the reviewer's reminder. We have provided the higher magnification of
microphotographs in the revised manuscript as follows (Figure R18).

**Figure R18. Representative immunofluorescence staining in experimental animals.**

Scale bars: 25µm.

*14) It is not clear from the introduction why the authors chose to explore childhood*
*asthma.*

Response:

Thanks for the reviewer's reminder. We have added this description in the Introduction
as follows:

"Allergic asthma is the most common clinical phenotypes of asthma. Notably, most
school-age children have allergic asthma, which has often obvious involvement in the
immune system such as eosinophils and type 2 helper T cells (Th2 cells) ⁶. Children
with allergic asthma have concomitant allergic sensitization, which has been associated
with asthma inception and severity".

*15) A concise hypothesis is missing.*

Response:

Thanks for the reviewer's reminder. We have added the concise hypothesis in the
discussion as follows:

"Collectively, this study highlights the critical role of METTL3 deficiency in the
pathogenesis of allergic asthma airway inflammation, as featured by promoting M2
macrophage activation and enhancing Th2 response, and uncovers a previously
unrecognized signaling axis involving METTL3/YTHDF3-m⁶A/PTX3/STX17 in
macrophage activation and autophagy maturation."

*16) Certain references are missing (i.e. line 87).*

Response:

Thanks for the reviewer's reminder. We have added these references to support this
conclusion ^{24, 25}.

*17) The manuscript would benefit from revision by an English native speaker.*

Response:

Thanks for your suggestion. We have tried our best to polish the language and also
involved native English speakers for language corrections in the revised manuscript.

**References**

- 1. Linder B, Grozhik AV, Olarerin-George AO, Meydan C, Mason CE, Jaffrey SR.
Single-nucleotide-resolution mapping of m6A and m6Am throughout the
transcriptome. *Nat Methods* **12**, 767-772 (2015).
- 2. Wang JN, *et al.* Inhibition of METTL3 attenuates renal injury and inflammation
by alleviating TAB3 m6A modifications via IGF2BP2-dependent mechanisms.
*Sci Transl Med* **14**, eabk2709 (2022).
- 3. Liu J, *et al.* Landscape and Regulation of m(6)A and m(6)Am Methylome across
Human and Mouse Tissues. *Mol Cell* **77**, 426-440 e426 (2020).
- 4. Xiao Y, Wang Y, Tang Q, Wei L, Zhang X, Jia G. An Elongation- and Ligation-
Based qPCR Amplification Method for the Radiolabeling-Free Detection of
Locus-Specific N(6) -Methyladenosine Modification. *Angew Chem Int Ed Engl*
**57**, 15995-16000 (2018).
- 5. Xu W, *et al.* METTL3 regulates heterochromatin in mouse embryonic stem cells.
*Nature* **591**, 317-321 (2021).
- 6. Lambrecht BN, Hammad H. The immunology of asthma. *Nat Immunol* **16**, 45-
56 (2015).
- 7. Zhang M, *et al.* Epithelial exosomal contactin-1 promotes monocyte-derived
dendritic cell-dominant T-cell responses in asthma. *J Allergy Clin Immunol* **148**,
1545-1558 (2021).
- 8. Saradna A, Do DC, Kumar S, Fu QL, Gao P. Macrophage polarization and
allergic asthma. *Transl Res* **191**, 1-14 (2018).
- 9. Biswas SK, Mantovani A. Macrophage plasticity and interaction with
lymphocyte subsets: cancer as a paradigm. *Nature Immunology* **11**, 889-896
(2010).
- 10. Staples KJ, Hinks TS, Ward JA, Gunn V, Smith C, Djukanovic R. Phenotypic
characterization of lung macrophages in asthmatic patients: overexpression of
CCL17. *J Allergy Clin Immunol* **130**, 1404-1412 (2012).

- 11. He J, *et al.* Fbxw7 increases CCL2/7 in CX3CR1hi macrophages to promote
intestinal inflammation. *J Clin Invest* **129**, 3877-3893 (2019).
- 12. Bain CC, MacDonald AS. The impact of the lung environment on macrophage
development, activation and function: diversity in the face of adversity. *Mucosal*
*Immunol* **15**, 223-234 (2022).
- 13. McQuattie-Pimentel AC, Budinger GRS, Ballinger MN. Monocyte-derived
Alveolar Macrophages: The Dark Side of Lung Repair? *Am J Respir Cell Mol*
*Biol* **58**, 5-6 (2018).
- 14. GINA. 2022 GINA Report, Global Strategy for Asthma Management and
Prevention. ([https://ginasthma.org/gina-reports/.](https://ginasthma.org/gina-reports/)) (2022).
- 15. Reddel HK, *et al.* An official American Thoracic Society/European Respiratory
Society statement: asthma control and exacerbations: standardizing endpoints
for clinical asthma trials and clinical practice. *Am J Respir Crit Care Med* **180**,
59-99 (2009).
- 16. Custovic A, Siddiqui S, Saglani S. Considering biomarkers in asthma disease
severity. *J Allergy Clin Immunol* **149**, 480-487 (2022).
- 17. Price DB, *et al.* Blood eosinophil count and prospective annual asthma disease
burden: a UK cohort study. *Lancet Respir Med* **3**, 849-858 (2015).
- 18. Pham TH, Damera G, Newbold P, Ranade K. Reductions in eosinophil
biomarkers by benralizumab in patients with asthma. *Respir Med* **111**, 21-29
(2016).
- 19. Shiratori H, *et al.* THP-1 and human peripheral blood mononuclear cell-derived
macrophages differ in their capacity to polarize in vitro. *Mol Immunol* **88**, 58-
68 (2017).
- 20. Chanput W, Mes JJ, Wichers HJ. THP-1 cell line: an in vitro cell model for
immune modulation approach. *Int Immunopharmacol* **23**, 37-45 (2014).
- 21. Levy DS, Kahana JA, Kumar R. AKT inhibitor, GSK690693, induces growth
inhibition and apoptosis in acute lymphoblastic leukemia cell lines. *Blood* **113**,
1723-1729 (2009).
- 22. Binnemars-Postma K, Bansal R, Storm G, Prakash J. Targeting the Stat6
pathway in tumor-associated macrophages reduces tumor growth and metastatic
niche formation in breast cancer. *FASEB J* **32**, 969-978 (2018).

- 23. Arranz A, *et al.* Akt1 and Akt2 protein kinases differentially contribute to
macrophage polarization. *Proc Natl Acad Sci U S A* **109**, 9517-9522 (2012).
- 24. Liu J, *et al.* N-6-methyladenosine of chromosome-associated regulatory RNA
regulates chromatin state and transcription. *Science* **367**, 580-586 (2020).
- 25. Yu R, Li Q, Feng Z, Cai L, Xu Q. m6A Reader YTHDF2 Regulates LPS-
Induced Inflammatory Response. *Int J Mol Sci* **20**, (2019).

REVIEWER COMMENTS

Reviewer #1 (Remarks to the Author):

The authors addressed most of my concerns by adding a substantial amount of new data. I still believe that the quality of the meRIP-seq datasets is sub-optimal but the validation of the m6A site in the 3'UTR of PTX3 by an independent approach is convincing enough to validate the model. I therefore support publication

Reviewer #2 (Remarks to the Author):

The authors have been quite responsive to the reviewers' comments. I will allow Reviewer's 1 and 3 to comment on the responses to their critiques. In the response to my critiques, the authors performed the requested experiments, but I am very concerned about the response to my third comment regarding the statistical analysis that the authors performed. They state "Thus, to make the result more convincible, we excluded these outliers, and recruited another fifteen new children with allergic asthma and ten healthy controls." This is certainly not what I was suggesting or would have recommended when I suggested about using a non-parametric test because the data was not normally distributed. No data should have been excluded in the analysis, certainly not the "outliers" as the authors term that data. In addition, the goal of data analysis is not to make the results "more convincible" but instead to accurately reflect the true biology. The addition of more study subjects to achieve greater statistical significance is also a major concern and I suggest that a biostatistician should review this manuscript.

Reviewer #4 (Remarks to the Author):

The authors provided extensive revision to the manuscript, adding additional data to reinforce and support key statements made in the manuscript. These new data sufficiently address the comments.

Additional Comments:

1) It is important to confirm that proper experimental controls for flow cytometry were

used. Was a live/dead exclusion marker utilized? If so, the methods section for flow cytometry data should be revised to reflect the use of proper controls. It should also include how the gating of intracellular cytokine markers were performed (ex. FMO controls). Additionally, the use of live/dead exclusion markers and Fc block should be added to the methods section. These items are very important for rigor and reproducibility of data.

Point-by-point responses

To Reviewer: 2

*The authors have been quite responsive to the reviewers' comments. I will allow*
*Reviewer's 1 and 3 to comment on the responses to their critiques. In the response to*
*my critiques, the authors performed the requested experiments, but I am very concerned*
*about the response to my third comment regarding the statistical analysis that the*
*authors performed. They state " Thus, to make the result more convincible, we*
*excluded these outliers, and recruited another fifteen new children with allergic asthma*
*and ten healthy controls. " This is certainly not what I was suggesting or would have*
*recommended when I suggested about using a non-parametric test because the data*
*was not normally distributed. No data should have been excluded in the analysis,*
*certainly not the " outliers " as the authors term that data. In addition, the goal of*
*data analysis is not to make the results " more convincible " but instead to*
*accurately reflect the true biology. The addition of more study subjects to achieve*
*greater statistical significance is also a major concern and I suggest that a*
*biostatistician should review this manuscript.*

Response:

We are really sorry for our misunderstanding the reviewer's previous comments "All
the correlation analyses should be performed using the appropriate statistical test and it
appears that these correlations may not be significant as they are being driven by a very
few values at the extremes of the data set in several of the analyses." In the previous
manuscript, after the data on childhood asthma were tested to be non-Gaussian
distribution and analyzed by Spearman correlation, we excluded the outliers to achieve
greater statistical significance, while this analysis couldn't accurately reflect the true
biology. We are really sorry for this mistake, and greatly appreciate the reviewer's
comments again.

Here, in statistical analysis, all results in the manuscript were tested for Gaussian
distribution and homogeneity variance. For data in Gaussian distribution and with

homogeneity variance, parametric test was used to analyze, such as independent t-test,
one or two-way ANOVA, etc. For data in non-Gaussian distribution, non-parametric
test was used to analyze, such as the Mann-Whitney test, Spearman correlation, etc. For
data in Gaussian distribution and without homogeneity variance, Welch's correction
was used. In the revised manuscript, we have amended these descriptions in the
statistical analysis. We also have presented the source data and information of statistical
analysis in the Source Data file.

Furthermore, we checked again the clinical information on 55 childhood asthma. The
data on childhood asthma were tested to be non-Gaussian distribution. To accurately
reflect the true biology, we re-performed the Spearman correlation analysis without
excluding the outliers (a total of 55 asthma patients). The results suggested that the
expression of *METTL3* in monocyte-derived macrophages, or the *PTX3* circulating
levels from 55 asthma patients slightly lowered the correlation with disease severity,
compared to the previous 50 asthma patients, nevertheless, this analysis didn't change
our original conclusion. Furthermore, we found this correlation in 55 childhood asthma
was relatively higher than that in the original 40 patients (Figure R1).

Thus, more childhood asthma patients are needed to confirm the findings reported
herein. However, since it is difficult to recruit more childhood asthma due to the limited
time and funding, we further added the limitation of this analysis in the revised
discussion as follows: "Importantly, large cohorts of childhood allergic asthma need to
be constructed, which evaluate the possibility of *METTL3* or *PTX3* levels as potential
biomarker for the diagnosis and assessment of childhood allergic asthma."

Here, we have renewed the data from 55 childhood asthma and amended the
descriptions in the revised manuscript. We also presented the source data of 55, 50, and
original 40 childhood asthma and information on statistical analysis in the Source Data
file.

**Figure R1. Clinical correlation between METTL3, PTX3 and disease severity in**
 **childhood allergic asthma. (a) Spearman correlation analysis of monocyte-derived**
 **macrophages *METTL3* expression, blood eosinophils number, FeNO, and %FEV₁**
 **levels in 55, 50, and original 40 childhood asthma, respectively. (b) Spearman**
 **correlation analysis between PTX3 protein levels and blood eosinophils numbers, or**
 **FeNO in 55, 50, and original 40 childhood asthma, respectively. Red represented the 5**

excluded patients and Blue represented the 14 added patients in the previous Figure.

To Reviewer: 4

*The authors provided extensive revision to the manuscript, adding additional data to*
*reinforce and support key statements made in the manuscript. These new data*
*sufficiently address the comments.*

Response:

We thank the reviewer for carefully reading our manuscript and appreciate the helpful
comments and critical questions. Based on these suggestions and questions, we have
made changes to the original manuscript to improve our manuscript. We hope that the
revised manuscript will be better for the readers to understand our points and finally
meet with your approval to get published.

*1) It is important to confirm that proper experimental controls for flow cytometry were*
*used. Was a live/dead exclusion marker utilized? If so, the methods section for flow*
*cytometry data should be revised to reflect the use of proper controls. It should also*
*include how the gating of intracellular cytokine markers were performed (ex. FMO*
*controls). Additionally, the use of live/dead exclusion markers and Fc block should be*
*added to the methods section. These items are very important for rigor and*
*reproducibility of data.*

Response:

We sincerely appreciate the reviewer's reminder. In flow cytometry, cells were stained
with Fixable Viability Stain 780 (FSV780, BD Biosciences) to identify viable cells,
which have less staining with the fixable viability dye. Then, the cells were incubated
with an anti-CD16/32 monoclonal antibody (eBioscience) to prevent the non-specific
binding of antibodies to Fc receptors on immune cells. For the gating of intracellular
cytokines markers, cells were stained with surface markers antibodies, fixed and
permeabilized, followed by incubated with isotype control and various cytokine
antibodies, respectively. The isotype control plot was used to set the gates of

intracellular cytokines (Figure R2).
Here, we have added the following description in the Methods: “Cells were stained with
Fixable Viability Stain 780 (BD Biosciences) and then incubated with an anti-CD16/32
monoclonal antibody (ebioscience).” “For the gating of intracellular cytokines markers,
cells were incubated with isotype control and various cytokine antibodies, respectively.
The isotype control plot was used to set the gates of intracellular cytokines.” We have
renewed the figure exemplifying the gating strategy in the revised Supplementary
Information.

**Figure R2. Gating strategies used for flow cytometry.**

REVIEWERS' COMMENTS

Reviewer #2 (Remarks to the Author):

The authors have now responded appropriately to my concerns about their statistical methods.

Reviewer #4 (Remarks to the Author):

Thank you for addressing all the comments.

We appreciate the helpful comments and constructive criticisms of the reviewers. Here,
we respond point-by-point to the different questions and concerns raised by the
reviewers.

Point-by-point responses

To Reviewer: 2

*The authors have now responded appropriately to my concerns about their statistical*
*methods.*

Response:

We sincerely appreciate the reviewer for the helpful comments on our study, and we
thank him/her for the positive response.

To Reviewer: 4

*Thank you for addressing all the comments.*

Response:

We sincerely appreciate the reviewer for the helpful comments on our study, and we
thank him/her for the positive response.
